# Thermochromic hydrogel with high transmittance modulation and fast response for flexible smart windows
Fan Jiang [1,4], Kui Yu [1,4] ✉, Roland Kieffer [1], Djanick de Jong [1], Richard M. Parker [2], Silvia Vignolini [2,3] & Marie-Eve Aubin-Tam [1] ✉

Growing environmental concerns are driving demand for energy-saving strategies. Thermochromic smart windows offer a practical solution by passively regulating sunlight in homes and offices. Despite recent progress, current technologies still face challenges in achieving the thermal durability and mechanical robustness necessary for long-term use, combined with a rapid transition below 30 °C. Here we report a thermochromic hydrogel assembled from poly(N,N-dimethylaminoethyl methacrylate) and 2,2,2-trifluoroethyl methacrylate that produces flexible films on a large scale. This hydrogel rapidly ( ~ 3 s) and reversibly becomes turbid above a tunable transition temperature spanning the human comfort zone, and maintains its thermochromic property even when mechanically stretched with 500% strain. The film's high modulation of solar transmittance (70.6%) and luminous transmittance (85.7%) enables efficient sunlight screening in hot weather and clear vision in cool weather. Such 'smart windows' remain stable for over 10,000 heating/cooling cycles. These combined features indicate the hydrogel suitability for applications ranging from heat-modulating smart windows (architectural, automotive, etc.) to passive temperature indicators and even wearables.

To tackle the ever-increasing global greenhouse effect and reach the goal of a carbon-neutral future, one crucial challenge is to decrease urban energy consumption via efficient thermal management of buildings[1,2]. As such, approaches like thermochromic smart windows, where the transmittance autonomously changes in response to temperature fluctuations, hold practical significance in reducing energy consumption arising from active air conditioning[3].

Previous attempts to develop thermochromic devices mostly focused on using vanadium dioxide ($VO_2$)[4–14] or poly(N-isopropylacrylamide) (PNIPAM) as the active element[15–21]. $VO_2$ undergoes a reversible metal-semiconductor transition at a critical temperature ($T_c$), resulting in the material being transparent below the $T_c$ while reflecting light above the $T_c$[4–6]. However, conventional thermochromic $VO_2$ films face several limitations, including a high $T_c$ (68 °C), low luminous transmittance ($T_{Lum}$) and low solar transmittance modulation ($\Delta T_{Sol}$). Strategies such as doping metal ions (e.g., Al, Mg, and W) in $VO_2$ films or fabricating nanothermochromics based on nanoparticles of $VO_2$ (doped with e.g., Zr, W, Ti, and Mg) can help to decrease $T_c$ and increase $\Delta T_{Sol}$ to a certain degree[7–14]. However, these

devices are generally unsuitable for most real-world applications, as both $T_{Lum}$ and $\Delta T_{Sol}$ must be maximized to ensure clear visibility and effective heat blockage. $VO_2$ systems with low transition temperatures are limited by their low $\Delta T_{Sol}$ of ~10%.

Hence, temperature-responsive polymers have emerged as a promising alternative; whereby light management could be pursued by dynamically altering the light-scattering behavior of the material. PNIPAM, is well known for its thermo-responsive behavior and, as such, has been widely investigated for smart window applications. It combines the advantage of a low transition temperature of ~32 °C, with a $T_{Lum}$ higher than 80% and a $\Delta T_{Sol}$ of ~50%[15]. Below its transition temperature, PNIPAM is miscible with water. However, above this threshold, hydrophobic interactions cause PNIPAM chains to collapse and expel water, which results in an increase in light scattering, hence decreasing the solar transmittance ($T_{Sol}$)[16]. The thermochromic performance of PNIPAM has been further improved by preparing it as a microgel suspension, hydrogel-derived liquid, or as a copolymer[15,17–23]. In addition to PNIPAM, other thermochromic systems, including hydroxypropyl cellulose (HPC) hydrogels[24], also demonstrate

[1]Department of Bionanoscience, Kavli institute of Nanoscience, Delft University of Technology, Van der Maasweg 9, Delft, The Netherlands. [2]Yusuf Hamied Department of Chemistry, University of Cambridge, Lensfield Road, Cambridge, United Kingdom. [3]Sustainable and Bio-inspired Materials, Max Planck Institute of Colloids and Interfaces, Potsdam, Germany. [4]These authors contributed equally: Fan Jiang, Kui Yu. ✉e-mail: yukui91@163.com; m.e.aubin-tam@tudelft.nl

high $\Delta T_{Sol}$, excellent durability, and/or tunable transition temperature. In most polymer-based thermochromic systems, rigid glass substrates serve as the standard configuration to encapsulate the thermochromic layer, owing to their excellent optical performance, high mechanical strength and long-term stability[25]. However, flowing liquid-based hydrogel systems might be confronted with potential leakage[26]. The PNIPAM hydrogel can also be prepared in the solid state and improved through various methods, such as copolymerizing with other monomers (acrylamide, acrylic acid)[27,28], adding cellulose derivatives (hydroxypropylmethylcellulose (HPMC), HPC)[29,30], and doping with silver nanorods[31]. Other organic solid-state thermochromic materials have likewise been studied, including HPMC-based hydrogel[32], HPC composite hydrogel[33], ionogels[34], and N,N-dimethylaminoethyl methacrylate-based (DMAEMA) hydrogels[35,36]. These solid-state thermochromic systems exhibit promising heat modulation and anti-leakage properties, but they also show limitations such as a slow response or a transition temperature outside the temperature comfort zone of the human body, i.e., between 20 and 30 °C[37] (see Supplementary Table 1 for a summary of the performance of existing technology). Moreover, the durability and thermal stability of many smart window systems have not been fully explored and optimized.

In this study, we address these challenges in thermochromic smart window design by developing a solid-state, and flexible material with thermal stability that exhibits a rapid and tunable thermochromic response within the human comfort temperature range. The material should ideally possess the advantages of polymer-based systems, such as low transition temperatures and high $\Delta T_{Sol}$, as well as the structural integrity of solid-state devices. Poly(DMAEMA) hydrogel is a thermochromic material with a transition temperature around 50 °C, tunable thermal responsiveness, and facile curing conditions[38], presenting huge potential and value for optimization. To further reduce the transition temperature and increase the response rate for practical applications, we copolymerized DMAEMA with hydrophobic fluorinated monomers. Owing to weakened water bonding and enhanced water repellency by the hydrophobic moieties, the copolymerized material experienced a more readily triggered phase separation, leading to a lower transition temperature and faster opacity change.

The specific thermochromic hydrogel system employed in this study is based on poly(DMAEMA/2,2,2-trifluoroethyl methacrylate (FMA)) (DMFM). As shown in Fig. 1, this DMFM hydrogel is transparent at room temperature (20 °C), but turns opaque at warmer temperatures. It has a fast response rate (~3 s), with a transition temperature tunable from 24 to 39 °C, which enables rapid blocking of the heat induced by solar irradiation, helping buildings maintain a temperature comfortable for the human body. By encapsulating this solid-state hydrogel with notable mechanical strength in a soft and transparent substrate, such as polyethylene terephthalate (PET), it can be fabricated into a flexible thermochromic device that is capable of operating for more than 10,000 heating/cooling cycles and 10 h of continuous operation without visible performance decay. The DMFM hydrogel was successfully integrated into a model house for heating/cooling simulation. Finally, we showcased that the same DMFM hydrogel can be used as a wearable, a temperature indicator, and an electricity-controlled device.

## Results and discussion
### Device fabrication and working mechanism
The thermochromic DMFM hydrogel was prepared through crosslinking DMEAMA and FMA monomers in the presence of water (Fig. 1a). Specifically, 57.9 vol% of DMAEMA, 5.2 vol% of FMA, and 36.9 vol% of deionized water were mixed together with a small amount of photoinitiator 2-hydroxy-4′-(2-hydroxyethoxy)−2-methylpropiophenone (Irgacure 2959; 0.05 wt%) and crosslinker N,N′-methylenebisacrylamide (MBA; 0.05 wt%). A device was then constructed by sandwiching this mixture between flexible PET sheets and curing with UV light at 365 nm wavelength (Fig. 1b), following a simple fabrication process (Supplementary Fig. 1). The resulting DMFM hydrogel device, called DMFM-4, is uniformly transparent at 20 °C (Fig. 1c), but becomes turbid when the temperature is increased to 30 °C

(Fig. 1d). The uncured DMFM-4 precursor solution also presents a reversible change in opacity in response to temperature variations (Supplementary Fig. 2). The responsiveness of the DMFM-4 hydrogel device to both temperature changes and mechanical stimuli is elegantly evidenced in Supplementary Movie 1, where there is a visible change in turbidity and reversible deformation in shape upon being held and bent by hand.

We hypothesize that the thermochromism property of the DMFM hydrogel is based on a reversible phase separation due to the formation/breaking of hydrogen bonds and a change of hydrophobicity. The lower transition temperature in comparison to poly(DMAEMA)[39] is attributed to the copolymerization with FMA, where the hydrophobic trifluoromethyl groups of FMA promote a more readily triggered phase separation. As schematically shown in Fig. 1e, f, at room temperature, DMFM polymer chains and water would be uniformly distributed, due to hydrogen bonds between hydrophilic tertiary amine groups of DMAEMA and water molecules[40]. As the temperature increases, the water/polymer hydrogen bonds would break, and the hydrophobic trifluoromethyl groups on FMA become dominant[38,41], resulting in the repulsion of water molecules and the formation of microscale water pockets inside the hydrogel due to phase separation[42]. Mie scattering of light is then expected to take place at the interface of these interlaced water pockets, resulting in an increase in turbidity.

The polymerization process was characterized by comparing the Fourier-transform infrared spectroscopy (FT-IR) spectra of uncured DMFM-4 solution and the cured DMFM-4 hydrogel (Supplementary Fig. 3a). The C=C stretching band at around 1637 cm$^{-1}$ and the C=C twisting bands at 815 and 944 cm$^{-1}$ found in the precursor solution experienced a decrease after being cured and crosslinked into a hydrogel (Supplementary Fig. 3b, c) indicating a reduction of C=C bonds in the sample[43]. In addition, the C=O stretching band found in the uncured solution at around 1713 cm$^{-1}$ demonstrated a shift to 1721 cm$^{-1}$ in the cured polymer, which corresponds to changes during radical polymerization from unsaturated ester to saturated ester groups[44].

### Thermochromic properties, durability, and tunability of materials
To quantify the thermochromic response of the DMFM-4 hydrogel, the solar transmittance spectra ($\lambda = 250$ to 2500 nm) of DMFM-4 with hydrogel thickness of 0.5 mm under different temperatures (21 to 48 °C) were measured (Fig. 2a). At 21 °C, the $T_{Sol}$, infrared transmittance ($T_{IR}$) and $T_{Lum}$ were $81.6 \pm 1.0\%$, $78.0 \pm 0.5\%$, and $85.7 \pm 1.5\%$, respectively, demonstrating high transparency in both visible and infrared regions at room temperature (Fig. 2b). When the temperature increased to 27 °C, the $T_{Sol}$, $T_{IR}$, and $T_{Lum}$ started to decrease to $30.1 \pm 2.1$, $38.0 \pm 3.6$, and $23.8 \pm 1.2\%$, respectively, indicating a transition temperature of DMFM-4 at around 27 °C. As the temperature further increased to 30 °C, the DMFM-4 hydrogel device became quite turbid with a low solar transmittance of $11.0 \pm 0.9\%$, leading to a high $\Delta T_{Sol}$ of $70.6 \pm 1.9\%$. Upon further increasing the temperature, the solar transmittance of the DMFM-4 hydrogel device continued to decrease slightly and reached $10.3 \pm 0.7\%$ at 48 °C. The PET films used to sandwich the hydrogel possess stable solar transmittance across this temperature range, measured as 77.1% at 20 °C and 77.4% at 35 °C (Supplementary Fig. 4a). Although glass sheets possess higher transparency, PET films were employed to endow the DMFM device with flexibility and low weight. It could be summarized that DMFM-4 underwent a rapid reduction in transparency from ~27 °C, with a stable minimum reached above ~30 °C.

To investigate the mechanism behind the thermochromic properties, confocal fluorescence microscopy was employed to characterize the DMFM-4 hydrogel at various temperatures. Due to the light scattering by the DMFM hydrogel under high temperatures, a limited amount of light can be transmitted through the Transmission Digital (TD) channel, appearing to be entirely black in the observed images. To solve this problem, a fluorescent dye was used, as previously reported to visualize elastic substrate deformations[45,46]. Similarly, 0.05 mg/mL of Rhodamine B was incorporated into the DMFM-4 hydrogel to monitor dynamic structural changes during phase transitions. As shown in Supplementary

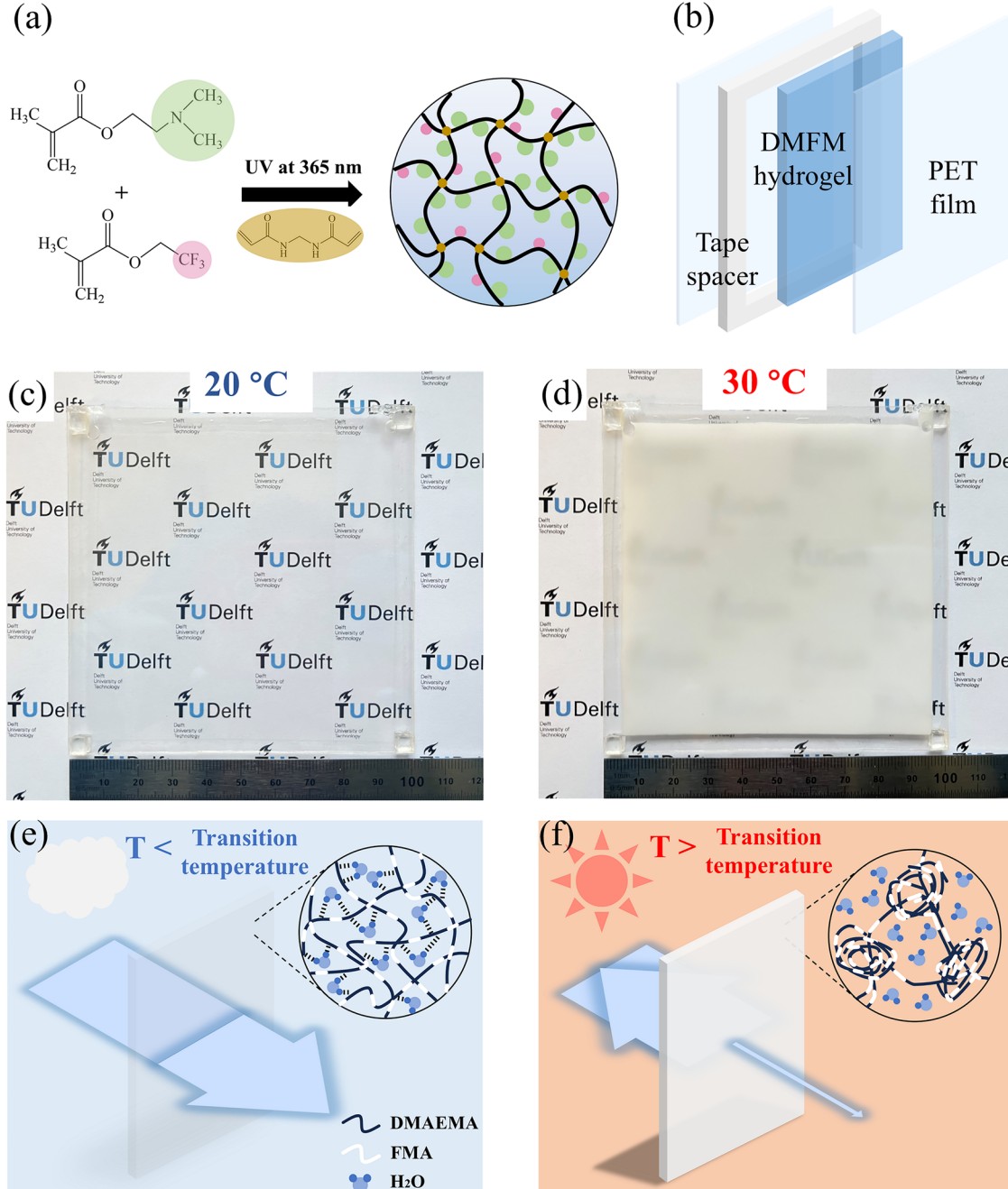

**Fig. 1 | Fabrication and transmittance modulation of DMFM hydrogel device.**
**a** Schematic of UV-polymerization and chemical structures of DMAEMA and FMA monomers. **b** A thermochromic device is constructed by laminating the DMFM hydrogel between PET sheets. **c**, **d** Images of the thermochromic device composed of

DMFM-4 hydrogel with dimensions of 10 cm × 10 cm × 0.5 mm showing a transparent state at 20 °C (**c**) and a turbid state at 30 °C (**d**). **e**, **f** Schematics of the proposed mechanism for the transparency change in the DMFM hydrogel device.

Fig. 5a, upon heating, a large number of black (non-fluorescent) microspheres emerged from the originally transparent and fluorescent hydrogel (Supplementary Fig. 5b). Rhodamine B is expected to remain in the polymer phase, and as such, the non-fluorescent voids in the images represent microscale water cavities that form during the phase separation process. These water pockets were found to have a cross-section ranging between ~400–1000 nm in size (Supplementary Fig. 5c). A similar experiment was also conducted with Nile Red, a hydrophobic dye that fluoresces in the organic phase but is quenched in the aqueous phase[47], again resulting in non-fluorescent microspheres emerging from the fluorescent hydrogel upon heating, further confirming that the spherical non-fluorescent voids correspond to water cavities (Supplementary Fig.

5d). Such inhomogeneity in the material explains the opacity of DMFM hydrogels under higher temperature.

To gain further insight into the transition temperature and reversible phase separation process, DSC analysis was performed to characterize the DMFM-4 hydrogel (Fig. 2c). During the heating process, hydrogen bonds with water can break, thus triggering a phase separation inside the DMFM-4 hydrogel. Accordingly, an endothermic transition was revealed at 27.3 °C, which aligns with the transition temperature (~27 °C) characterized by ultraviolet/visible/near infrared (UV/Vis/NIR) spectroscopy. Furthermore, an exothermic transition at 26.4 °C was also observed during the cooling process of the same sample, which is again consistent with the DMFM hydrogel undergoing a reversible recovery from the phase separation. The

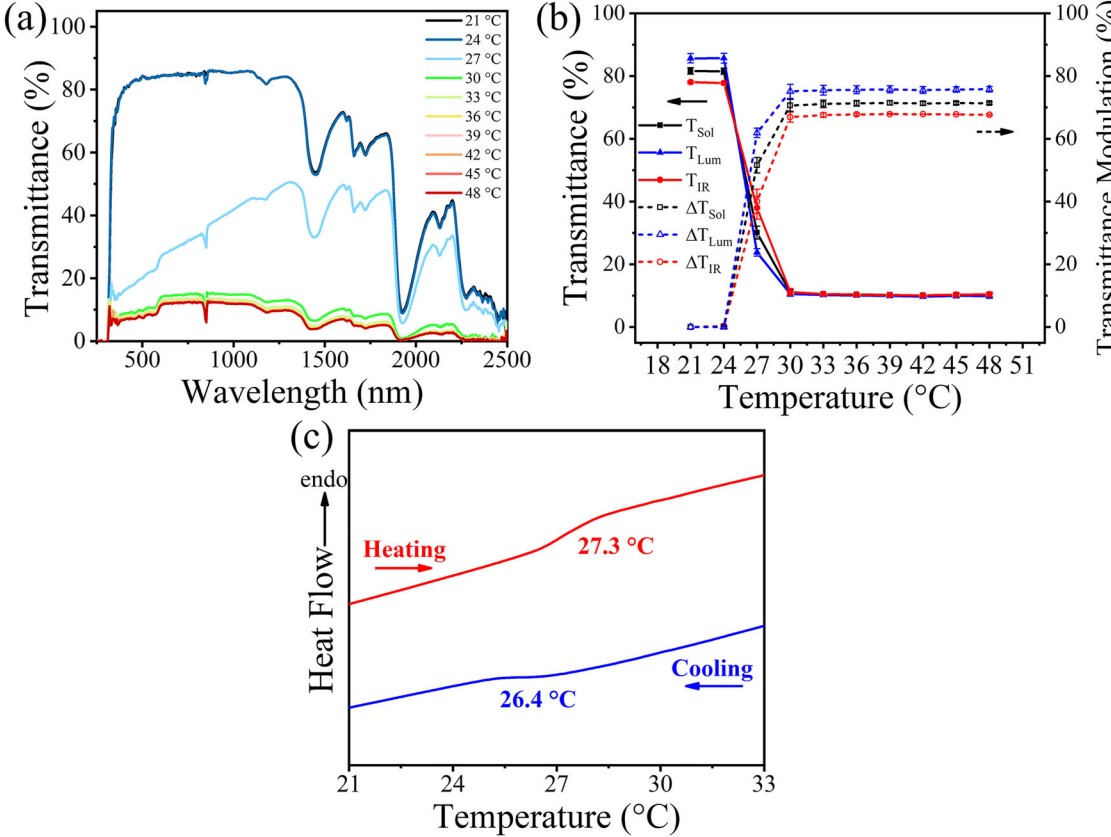

**Fig. 2 | Thermochromic behavior and transition temperature of DMFM-4 hydrogel device. a** Solar transmittance spectra (250 to 2500 nm) of the DMFM-4 hydrogel device with a layer thickness of 0.5 mm upon temperature modulation from 21 to 48 °C, showing that the transmittance change occurs at 27 °C. **b** $T_{Lum}$, $T_{IR}$, and $T_{Sol}$ (solid lines), and corresponding luminous transmittance modulation ($\Delta T_{Lum}$), infrared transmittance modulation ($\Delta T_{IR}$), and $\Delta T_{Sol}$ (dashed lines) under temperatures from 21 to 48 °C with error bars indicating standard deviation ($N = 3$ DMFM-4 hydrogel devices). **c** Differential scanning calorimetry (DSC) of DMFM-4 hydrogel.

phase transition characterized in DSC also resembles that of other thermochromic materials based on phase separation[48,49]. The thermal stability and water loss profile of the DMFM-4 hydrogel was also evaluated by thermogravimetric analysis (TGA) (Supplementary Fig. 6), showing the value of encapsulating the hydrogel.

To explore the durability of DMFM-4 hydrogel smart windows, a cyclic test was performed by repeatedly cooling to 20 °C and heating to 35 °C using alternating cool and hot water flows, with each cycle taking ~20 s (Supplementary Fig. 7a). Figure 3b shows that the lowest and highest optical transmittance of DMFM-4 remains at stable values during the 10,000 heating/cooling cycles, demonstrating excellent cycling stability. The transmittance curve during the first three and the last three heating-cooling cycles is magnified and depicted in Fig. 3a, displaying a decay of transmittance modulation of less than 4% after 10,000 repeated thermochromic cycles. The DMFM-4 device remained transparent and homogeneous after 10,000 cycles and could still demonstrate opacity change under higher temperature (Supplementary Fig. 8). Moreover, to test the stability of DMFM-4, a continuous 10-h period at warm (35 °C) or cold (20 °C) temperatures was applied (Fig. 3c). The transmittance of the DMFM-4 hydrogel device at 660 nm remained stable with transmittance modulation of 89.1 ± 0.8% over the 10-h measurement. The temporal response of the DMFM-4 hydrogel device was measured and characterized (Fig. 3a). The DMFM-4 thermochromic device turned transparent in ~3 s in cool water and then turned turbid in around 3 s under hot water flow. Nearly immediate coloration and bleaching events corresponding with external temperature change could be observed on the DMFM-4 hydrogel smart window. Slow heating from 20 to 31 °C over 12 min was also performed on the

DMFM-4 hydrogel device to investigate the transition and saturation temperatures (Supplementary Fig. 7b).

As the thermochromic behavior of DMFM relies on the reversible phase separation between water and polymer chains, changing the proportion of water in the DMFM hydrogels is likely to tune the transition temperature. To assess this, DMFM hydrogel devices with different water concentrations were fabricated: ranging from 18.9 vol% for DMFM-1 up to 41.2 vol% for DMFM-5. The transmittance spectra ($\lambda = 250–2500$ nm) of these samples were measured for different temperatures (Supplementary Fig. 9), and the $T_{Sol}$ were calculated and compared (Fig. 4a). The transition temperatures of DMFM-1, DMFM-2, DMFM-3, DMFM-4, and DMFM-5 were respectively measured to be ~39, ~33, ~30, ~27, and ~24 °C. DSC analyses of the DMFM hydrogels with different water concentrations also indicated that a decrease of water content causes the transition temperature to shift toward a higher temperature (Fig. 4b). In accordance with these results, optical microscope images of the DMFM hydrogels at various temperatures (Fig. 4c) show that the transition temperature increased as the water concentration decreased in hydrogel devices from DMFM-4 to DMFM-1; while the DMFM-5 hydrogel device was already slightly turbid at room temperature. As opposed to PNIPAM, the transition temperatures of DMFM hydrogels relate to the water concentration. We hypothesize that this originates from the protonation of tertiary amine groups and the corresponding change of charge density in the hydrogel[50,51], which affects the hydrogen bonding network and polymer hydrophilicity[42,52]. In addition, the polymer chains with hydrophobic trifluoromethyl groups could aggregate more easily in a water-rich hydrogel[53]. The tunability of the transition temperature reveals the high potential of this

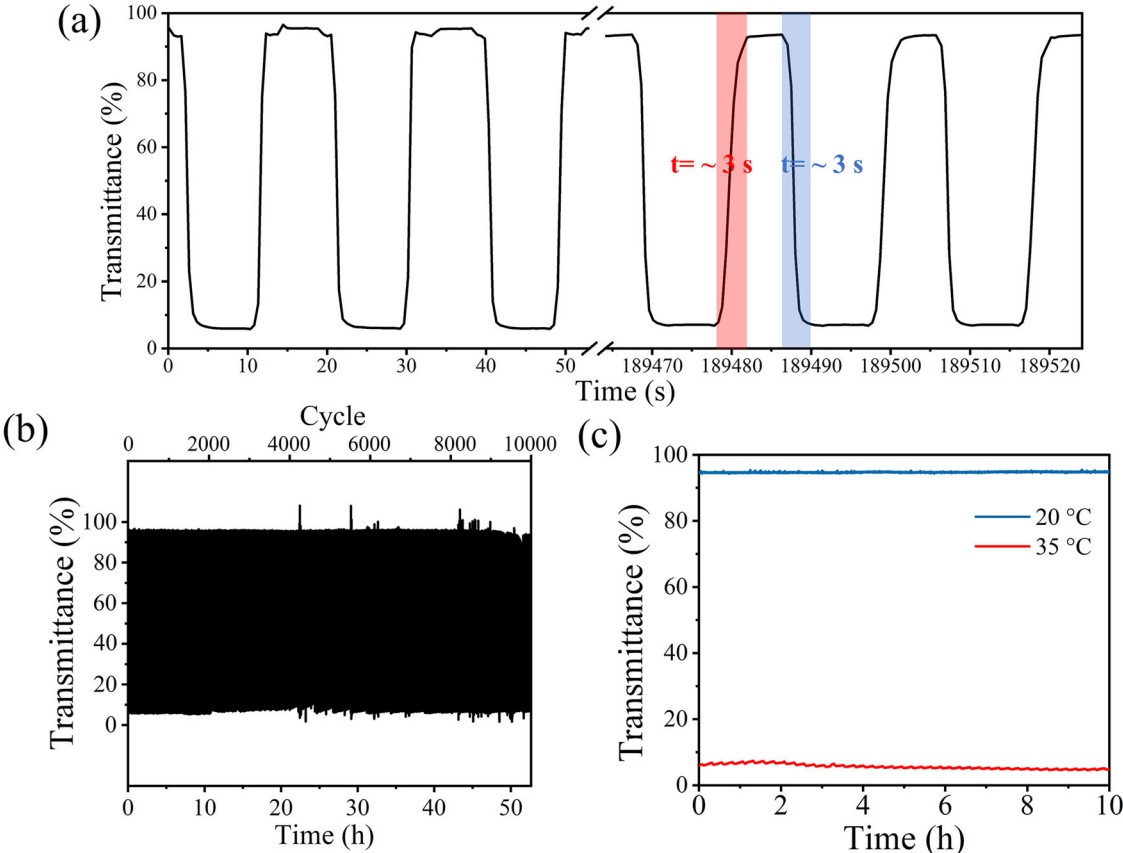

**Fig. 3 | Cyclic stability and durability of the DMFM-4 hydrogel device.**
**a** Magnified curve of the dynamic transmittance spectrum of the DMFM-4 hydrogel device during the first three and the last three cycles (after 52.6 h). **b** Dynamic transmittance spectrum of DMFM-4 hydrogel device over 10,000 heating/cooling cycles (52.6 h). **c** Dynamic transmittance spectrum of DMFM-4 hydrogel device at 660 nm under 20 and 35 °C for 10 h.

thermochromic hydrogel device for diverse application scenarios, for example, DMFM-2 (transition temperature ~33 °C) for tropical climates and DMFM-4 (transition temperature ~27 °C) for temperate climates.

### Demonstration of potential applications

To evaluate the temperature modulation performance of DMFM-4 smart windows in realistic atmospheric conditions, a model house with external infrared (IR) heating and internal temperature monitoring was built, as shown in Fig. 5a. The model house was made with both plywood and Styrofoam to better simulate the heat insulation property of real-life houses. When the indoor temperature was comfortable (around 25 °C), the DMFM smart window remained transparent, allowing light to pass through and illuminate the interior. However, when the indoor temperature exceeded the human comfort zone, the hydrogel device autonomously became turbid and blocked light from outside to prevent further increases in the indoor temperature. Figure 5b shows the temperature monitoring inside the model house equipped with different window materials during light on and off periods. The indoor temperatures started at ~22.5 °C, and after 30 min of IR irradiation (780–1400 nm, peaked at ~1000 nm), the temperature inside the house with only a glass window and with a glass window with empty PET films increased sharply to 38.8 ± 0.5 °C and 38.2 ± 0.1 °C, respectively. In contrast, the DMFM-4 smart window managed to keep the temperature inside the model house at 34.0 ± 0.3 °C, i.e., 4.8 °C lower than the bare glass window. After the light was turned off, the inside temperature dropped down to ~24 °C within a similar time period of 15 min for all test groups. Tests were also performed on a model house outdoors on a sunny day, confirming that DMFM-4 smart windows led to reduced heating compared to glass windows (Supplementary Fig. 10). These observations indicated that the DMFM thermochromic device can act as a dynamic and autonomous temperature modulation system.

Furthermore, due to the easy and cost-effective 'stir-inject-cure' fabrication process, the thermochromic laminate can be readily scaled up. To demonstrate this, an A4-size PET thermochromic device with DMFM-4 hydrogel inside was fabricated (Fig. 5d). The flexibility of DMFM smart windows, owing to the DMFM hydrogel and the great flexibility of the PET substrates, are displayed in Fig. 5e and Supplementary Movie 1. The free-standing solid-state DMFM-4 hydrogel (without PET) exhibited a tensile strength of 56.6 ± 3.8 kPa and high stretchability with a strain at break of 532.5 ± 52.2% during tensile testing (Supplementary Fig. 11). Notably, the material still displayed thermochromic behavior when stretched by 500%, without showing obvious shrinkage or embrittlement above the transition temperature. The combination of good mechanical properties and thermal stability over 10,000 cycles indicates an excellent durability for long-term operation. This durability could be attributed to several factors: Firstly, the encapsulation within PET effectively protects the hydrogel from direct mechanical damage, leading to a longer lifetime during thermal cycles. Secondly, the polymerized solid-state hydrogel exhibits good mechanical strength and structural robustness, which helps to maintain its structure and performance under repeated thermal stress. We found that after multiple drops from a 1-m height and subsequent bending to 90°, no visible leakage or structural damage was observed in the PET-encapsulated DMFM-4 hydrogel device (Supplementary Fig. 12a). The unchanged weight of the device after dropping and bending further confirmed the anti-leakage property (Supplementary Fig. 12b), owing to the encapsulation within PET and the mechanical strength of DMFM hydrogel. Thirdly, we expect that the crosslinking of the hydrogel helps to construct a more uniform and well-aligned polymer chain network, further enhancing the reversibility of phase separation in the DMFM hydrogel.

The DMFM-4/PET laminate can be easily installed on existing glass windows without any modification (Fig. 5), due to its lightweight and

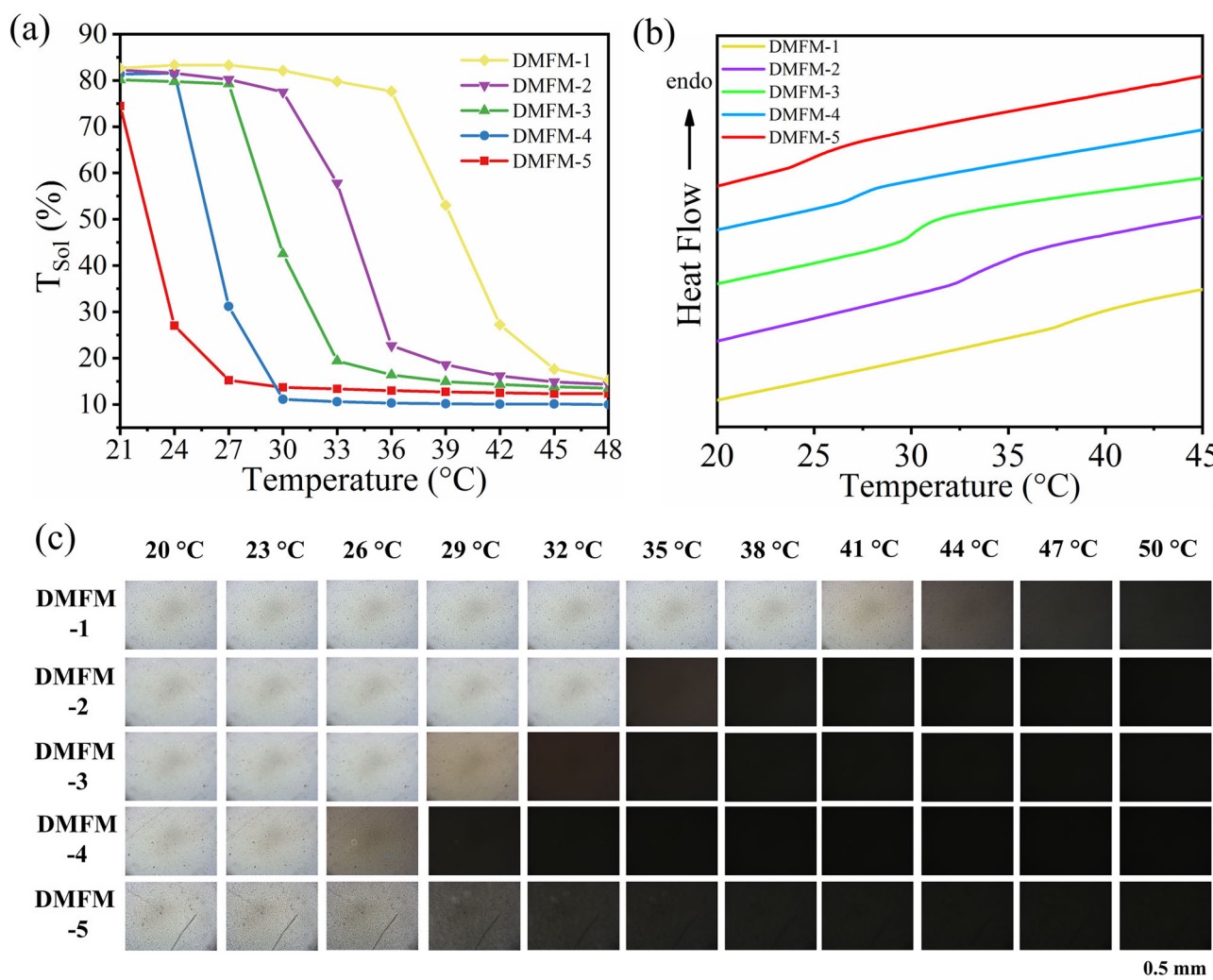

**Fig. 4 | Tunability of the thermochromic behavior of DMFM hydrogels. a** $T_{Sol}$ of DMFM hydrogel devices with different water concentrations in response to temperatures from 21 to 48 °C. **b** DSC of DMFM-1, DMFM-2, DMFM-3, DMFM-4, and DMFM-5 hydrogels. **c** Optical microscope images (transmission mode) of devices with different water concentrations, with a layer thickness of 0.5 mm at temperatures from 20 to 50 °C. A white appearance means that the hydrogel device is transparent and that the light can travel to the camera. When the hydrogel device becomes turbid, the light is not ballistically transmitted through the hydrogel, resulting in a black appearance.

excellent leak-proof characteristics. Furthermore, the flexibility of this thermochromic device enables it to be employed on curved glass, expanding the application scenarios from houses to vehicles, or even wearables. As shown in Supplementary Fig. 13, an autonomous sun-blocking smart hat was fabricated by seamlessly embedding a flexible DMFM-4 hydrogel device into the curved brim of a baseball cap, which was customized with the color purple by using 0.01 wt% Rhodamine B to demonstrate the possibility of producing wearables in various colors. The hat brim becomes opaque under high temperatures and blocks sunlight from the wearer, but turns transparent under cooler or low-light conditions to enhance upward visibility when shading is unnecessary. This demonstration is a simple and battery-free solution for passive sun shading and real-time temperature indication, which can be incorporated into outdoor clothing, sports gear, fashion-related accessories, or the shades of outdoor terraces.

In addition to passive heat modulation, other actively controlled applications for DMFM hydrogels were also explored. To trigger the opaque state "on demand", an electrically controlled DMFM-4 hydrogel device was fabricated on a flexible and transparent indium tin oxide (ITO)-PET substrate (Fig. 6a and Supplementary Fig. 4b), with copper tapes used for connection to an external circuit. Since the ITO can produce heat when connected to a power source (Joule's Law)[54], the conductive substrate with

elevated temperature can then switch the thermochromic hydrogel into the turbid state. In Fig. 6a and Supplementary Movie 2, this bendable smart window stayed transparent in the OFF state (0 V), and turned turbid uniformly in the whole device area after switching to the ON state (7.0 V), using 0.56 W of electrical power. From the spectra of the DMFM-4 ITO-PET device (Fig. 6b), luminous transmittance ($\Delta T_{Lum} = T_{0 V} - T_{7.0 V}$) as high as $53.2 \pm 2.3\%$ could be attained at 560 nm. Therefore, by incorporating DMFM into ITO-PET films, the transparency can be controlled electrically.

Due to the intrinsic tunability of the transition temperature of the DMFM hydrogels obtained by varying the water concentration, patterned hydrogel systems (containing several types of DMFM hydrogels) can be built to achieve multi-stage thermochromism. As shown in Fig. 6c, a pixel-style TUD flame logo was designed and utilized on a DMFM patterned hydrogel system, in which TUD letters were made with DMFM-4 hydrogel (transition temperature ~27 °C), the flame was achieved with DMFM-2 hydrogel (transition temperature ~33 °C), while the rest was filled with DMFM-1 hydrogel (transition temperature ~39 °C). We found that this patterned hydrogel system was transparent and uniform at room temperature, but as the temperature elevated to 30 °C, the "TUD" letters appeared, followed by the flame pattern at 35 °C. The "TUD" flame pattern was clear and obvious until 40 °C, and gradually blended into the

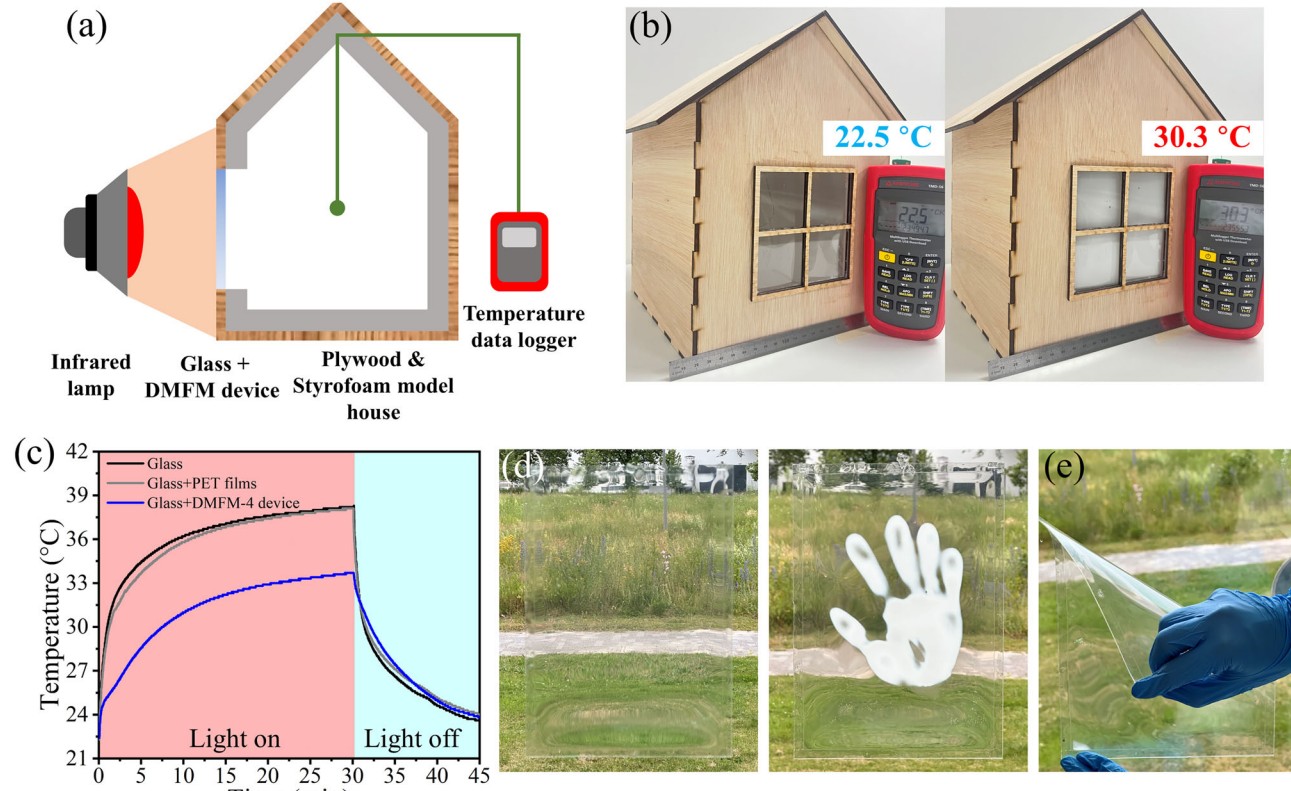

**Fig. 5 | Demonstration of heat modulation by the DMFM-4 hydrogel device for smart window applications. a** Schematic illustration of the experimental setup for monitoring the internal temperature of a model house. The dimensions of the model house are 21 cm × 21 cm × 28 cm, the thickness of plywood is 0.4 cm, the thickness of Styrofoam is 0.9 cm, and the glass window has dimensions of 10 cm × 10 cm × 0.1 cm. The distance between the infrared lamp and the house is 18 cm. **b** Photographs of the model house with DMFM-4 hydrogel device (dimensions: 11 cm × 11 cm × 0.5 mm)

directly mounted on the glass window before and after infrared irradiation. **c** Temperature profiles of a thermometer inside the model house under infrared irradiation with a glass window or empty PET frame on the glass window or DMFM-4 hydrogel device on the glass window ($N = 3$ measurements with DMFM-4 hydrogel device, data represent one of the three measurements). **d** Optical images of an A4-size DMFM-4 hydrogel device at transparent (left) and turbid (right) states triggered by hand with palm print. **e** The DMFM-4 hydrogel device is easily bent.

background and disappeared as the temperature increased to 45 °C. This strategy moves beyond a simple binary response to enable granular temperature indication for smart labeling, as well as an information encryption system that presents selected information at designated temperatures. Compared with liquid crystal thermometers that rely on delicate liquid crystal structures[55], the DMFM hydrogel shows higher optical contrast and resistance to physical damage.

## Conclusions

In summary, we have developed a solid-state flexible thermochromic DMFM hydrogel device with tunable transition temperature that shows an excellent solar transmittance modulation of 70.6% and luminous transparency of 85.7%. The rapid transparency change of PET-enclosed devices through simple contact with body temperature, combined with their scalability, highlights their prospects for practical applications. The transition temperature could be simply tuned from 24 to 39 °C by adjusting the water content of the DMFM hydrogel, without the need for unfavorable acid and alkaline chemicals employed in HPC hydrogels[24]. The hydrogel with a transition temperature of 27 °C showed a response time of 3 s and maintained its thermochromic response over 10,000 cycles. Compared with recently reported thermochromic devices (Fig. 7), the DMFM hydrogel smart window performs competitive $T_{Lum}$ and $\Delta T_{Sol}$, and shows a low transition temperature that coordinates with the human comfort zone. This was validated further by simulation of their heat insulation properties under real-world conditions.

The DMFM hydrogel exhibits great mechanical flexibility, thermal stability, and durable thermochromic performance, highlighting its

potential for long-term application in smart window systems. In addition, the DMFM hydrogel can be sandwiched between lightweight and flexible PET films. Therefore, DMFM thermochromic devices can easily be installed onto existing glass surfaces of buildings or curved sunroofs in vehicles to form smart windows. While glass-based smart windows remain the standard configuration due to their excellent performance and durability, our approach offers a complementary strategy that broadens the scope of application scenarios. These DMFM hydrogel devices also present high potential for wearables, temperature labels and information encryption.

## Experimental methodology

**Materials.** *N,N*-dimethylaminoethyl methacrylate (98%, Sigma-Aldrich), 2,2,2-trifluoroethyl methacrylate (99%, Sigma-Aldrich), *N,N′*-methylenebisacrylamide (Sigma-Aldrich), 2-hydroxy-4′-(2-hydroxyethoxy)-2-methylpropiophenone (98%, Sigma-Aldrich), Rhodamine B (≥95%, Sigma-Aldrich), Nile Red (Sigma-Aldrich), PET films (thickness: 0.1 mm), double-sided acrylic foam tape (tesa® ACX^plus 7054, 0.5 mm × 6 mm), double-sided acrylic foam tape (3M™ VHB™, 1 mm × 5 mm), ITO glass, ITO-PET films, cover glass (thickness: 170-μm, No. 1.5H), copper tape, silver conductive paint (Electrolube), infrared lamp (Greensen, 150 W). All the reagents were used as received.

**Preparation of DMFM precursor solutions and devices.** The DMFM precursor solutions were prepared through a simple mixing and stirring process, and the thermochromic devices were then fabricated as illustrated in Supplementary Fig. 1. First, 57.9 vol% of DMAEMA, 5.2 vol% of FMA and 36.9 vol% of DI water were mixed together, then 0.05 wt% of

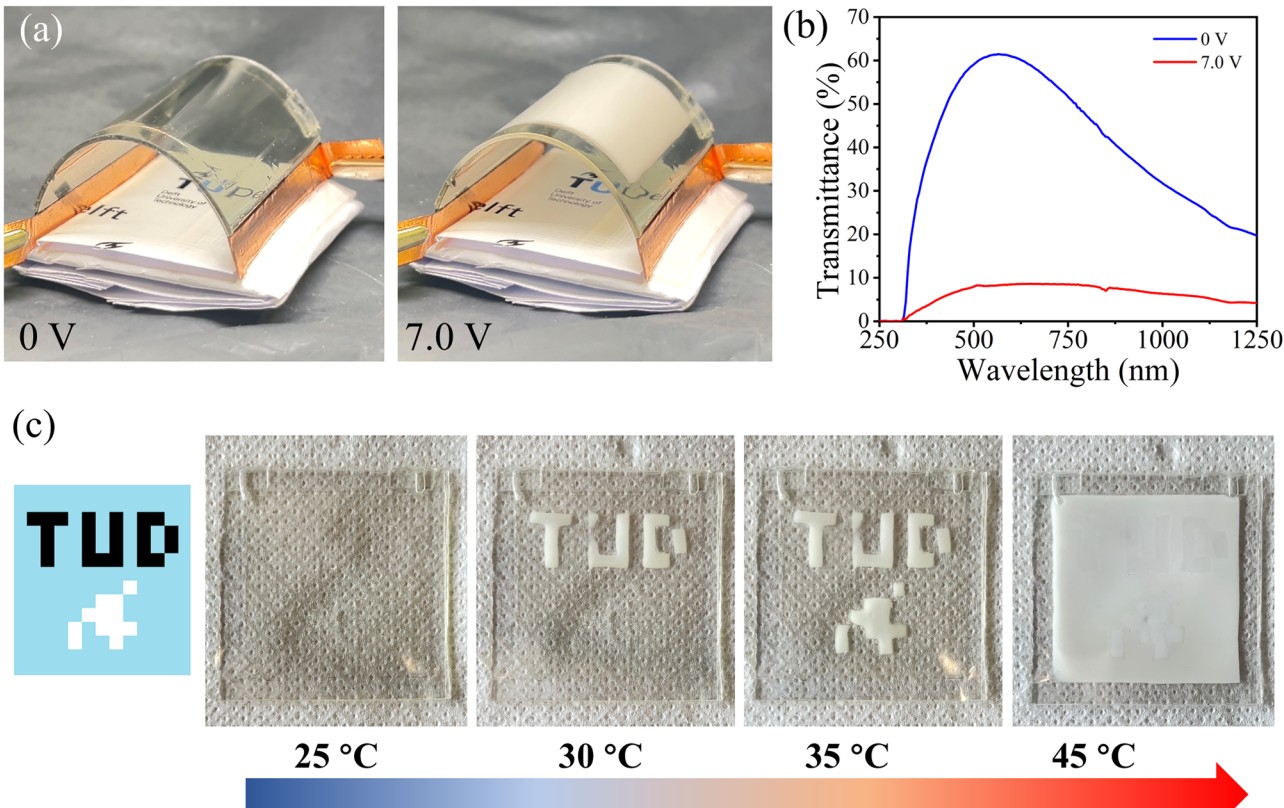

**Fig. 6 | Demonstration of active opacity change and pattern display. a** Optical images of electrically controlled flexible DMFM-4 hydrogel device fabricated on ITO-PET/PET films when either no voltage (left) or a voltage of 7.0 V (right) was applied (dimension: 5 cm × 4 cm × 0.5 mm). **b** Optical transmittance of DMFM-4 hydrogel device with a layer thickness of 0.5 mm when either no voltage (blue) or a voltage of 7.0 V (red) was applied (N = 3 DMFM-4 hydrogel ITO-PET/PET devices, data represent one of the three samples). **c** Process of displaying patterns in the hydrogel under increasing temperature with DMFM-1/2/4 hybrid hydrogel device (dimension: 40 mm × 40 mm × 1 mm).

Irgacure 2959 and 0.05 wt% of MBA were added into the solution and stirred for 5 min to form the precursor solution of the DMFM-4 hydrogel. The precursor solutions of DMFM-1 (74.3 vol% of DMAEMA, 6.8 vol% of FMA and 18.9 vol% of DI water), DMFM-2 (67.9 vol% of DMAEMA, 6.1 vol% of FMA and 26.0 vol% of DI water), DMFM-3 (62.5 vol% of DMAEMA, 5.7 vol% of FMA and 31.8 vol% of DI water), and DMFM-5 (53.9 vol% of DMAEMA, 4.9 vol% of FMA and 41.2 vol% of DI water) were prepared in a similar way. Then, the precursor solution was cured under UV light (365 nm, 34.2 mW/cm²) for 3 min in order to increase the viscosity of the precursor solution. Next, the precursor solution was injected between two transparent PET films, spaced by 0.5 or 1 mm with double-sided acrylic tape. Finally, the device was UV-cured for 20 min. For the DMFM-4 hydrogel devices characterized by confocal micro-scopy, glass coverslips with a thickness of 170 μm were used as substrates instead of PET films. The electrically controlled DMFM-4 hydrogel device was prepared similarly, but by sandwiching the hydrogel between one PET film (dimension: 4 cm × 4 cm × 0.1 mm) and one ITO-PET film (dimension: 5 cm × 4 cm × 0.1 mm). Conductive copper tape was wrap-ped on the left and right edges of the ITO-PET layer to serve as electrodes, with a certain length exposed as lead-outs for electrical connections. The measured electrical resistance of this device was 87.9 Ω.

**Characterization of a thermochromic hydrogel device.** UV/Vis/NIR Spectrophotometer (PerkinElmer LAMBDA 1050+ with integrating sphere, 250 to 2500 nm) was used to obtain UV/Vis/NIR spectra of samples. ITO glass, silver paste, copper electrodes and a DC power source were used to heat and control sample temperatures during measure-ments. A heating chamber (Linkam Scientific, THMS600) was employed to control the sample temperatures during measurements. Temperature-controlled transmission microscopy was carried out by mounting this heating stage to a customized Zeiss Axio Scope A1 microscope. The numerical aperture (NA) of the illumination condenser was 0.6. The sample was imaged using a combination of an EC Epiplan-Neofluar objective (5x NA0.13) and a CMOS camera (Eye IDS, UI-3580LE-C-HQ, white balanced against the empty Linkham stage). Image magnification was verified using a microscope slide scale bar. For the durability tests, we assembled and used a system with a power supply (Voltcraft ESP-3005S), digital thermometer (Amprobe TMD-56), water cooling/heating system with transmittance measurement (AlGaInP LED (Kingbright, 660 nm), thermocouple amplifiers (MCP9600), temperature data logger (PT-100, Pico Technology), digital light sensor (Adafruit TSL2591) and board (Arduino Beetle). FT-IR spectra were collected in the wavenumber range of 4000–600 cm⁻¹ using an FT-IR spectrometer (PerkinElmer Spectrum 100), which averaged over eight scans at a resolution of 2 cm⁻¹.

In order to calculate the $T_{Sol}$ (from 250 to 2500 nm), $T_{Lum}$ (from 380 to 780 nm), and $T_{IR}$ (from 780 to 2500 nm), Eq. (1) was used:

$$T_{Sol/Lum/IR} = \int \varphi_{Sol/Lum/IR}(\lambda) * T(\lambda)d(\lambda) / \int \varphi_{Sol/Lum/IR}(\lambda)d(\lambda) \quad (1)$$

where $\lambda$ is the wavelength, $T(\lambda)$ represents the measured transmittance at a specific wavelength, $\varphi_{Lum}(\lambda)$ stands for standard luminous efficiency function of photopic vision from 380 to 780 nm[56], and $\varphi_{Sol}(\lambda)$ and $\varphi_{IR}(\lambda)$ are respectively the solar and infrared irradiance spectrum for air mass 1.5[57]. $\Delta T_{Sol/Lum/IR}$ is calculated through Eq. (2) as follow:

$$\Delta T_{Sol/Lum/IR} = T_{Sol/Lum/IR, 20\,°C} - T_{Sol/Lum/IR, 30\,°C} \quad (2)$$

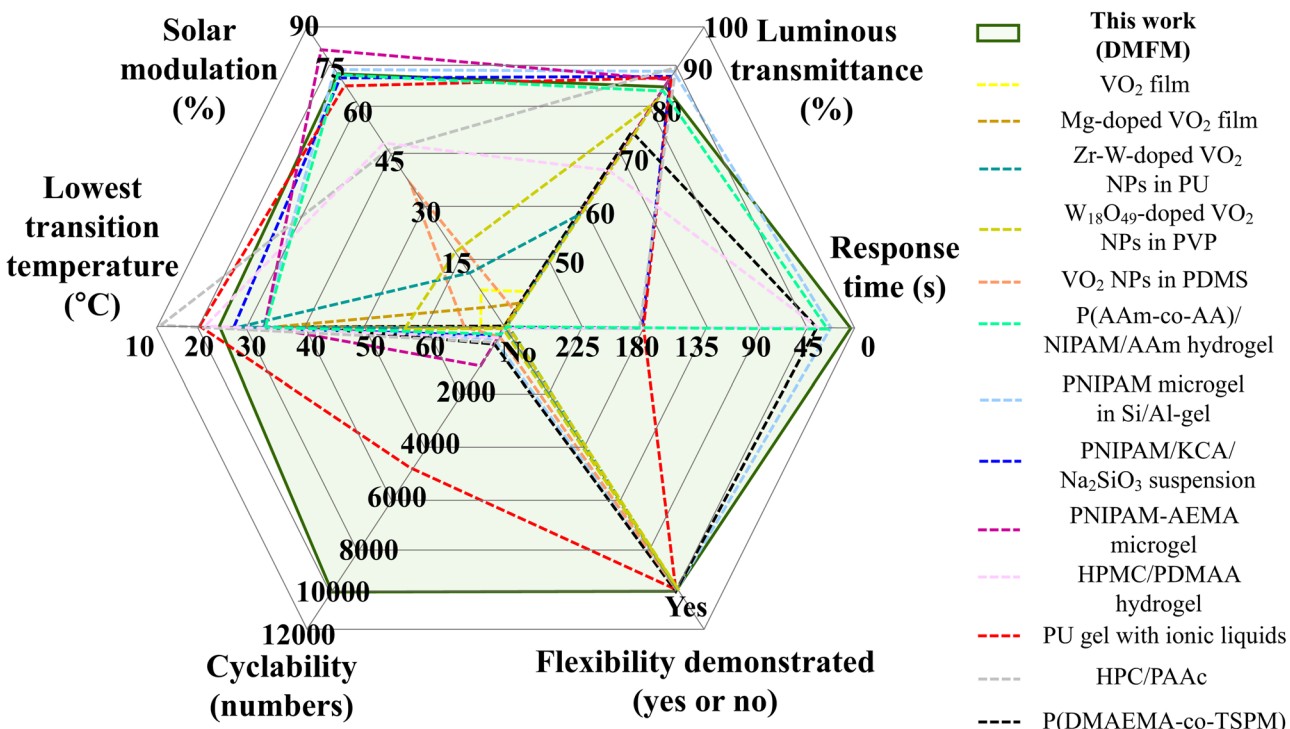

**Fig. 7 | Performance comparison against previously reported thermochromic devices.** Thermochromic performances (solar modulation, luminous transmittance, lowest transition temperature, cyclability, response time, and demonstrated flexibility) of DMFM hydrogel compared with previously reported work (see Supplementary Table 1 for details)[6,8,10,13,14,17,18,20,24,27,32,34,36].

A confocal microscope (Nikon A1plus Confocal equipped with Plan Apo IR 60x WI DIC N2 objective (NA = 1.27)) with a 561 nm excitation laser and a 593/46 nm bandpass emission filter (third filter cube) was used to assess the mechanism of transparency change. During acquisition, 4x optical zoom and 4x line averaging were applied to enhance spatial resolution and reduce noise. Fast imaging was employed to observe live and dynamic changes in the samples. To capture clear details and achieve high resolution, the hydrogel was imaged before and during heating at a rate of 7.5 frames/second, then the frames were stacked together by summing individual frames to increase contrast. Prior to particle size analysis, an intensity threshold was applied to the summed frames in ImageJ, followed by image binarization. Particle analysis was then performed using the "Analyze Particles" function in ImageJ, which determined the area of each water cavity based on the count of white pixels (a 156 pixel × 145 pixel wide view of a region of interest that is 16.15 × 15.01 μm in size). Differential scanning calorimetry (PerkinElmer Diamond DSC) was used to determine the transition of the hydrogel by cooling/heating at 2 °C min$^{-1}$ from 20 to 45 °C. Thermal stability of the hydrogel was determined by thermogravimetric analysis (PerkinElmer TGA 8000) through ramping the temperature from 30 to 700 °C at 10 °C min$^{-1}$ in air. Tensile tests were performed by an electromechanical universal testing machine (Instron 34SC-05) at 10 mm min$^{-1}$ with sample dimensions of 30 mm × 9 mm × 1 mm.

## Data availability

The data that support the findings of this study are available in the 4TU.ResearchData repository at https://doi.org/10.4121/87510030-e06a-42fc-831b-9b8e01470bca.

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

## Acknowledgements
We want to thank Dr. Rienk Eelkema for the useful discussion. The authors are supported by the European Research Council (ERC starting grant no. 101042612) and Engineering and Physical Sciences Research Council (EPSRC Program Grant Value EP/W031019/1).

## Author contributions
F. J. (Conceptualization: Lead; Data curation: Lead; Formal analysis: Lead; Investigation: Lead; Methodology: Lead; Project administration: Equal; Visualization: Lead; Writing—original draft: Lead; Writing—review & editing: Equal). K.Y. (Conceptualization: Lead; Formal analysis: Lead; Investigation: Lead; Methodology: Lead; Supervision: Equal; Writing—original draft: Lead; Writing—review & editing: Equal). R.K. (Data curation: Supporting; Formal analysis: Supporting; Investigation: Supporting; Methodology: Supporting; Project administration: Supporting; Resources: Equal; Writing—review & editing: Supporting). D.d.J. (Investigation: Supporting; Methodology: Supporting; Writing—review& editing: Supporting). R.M.P. (Investigation: Supporting; Methodology: Equal; Resources: Supporting; Writing—review & editing: Equal). S.V. (Investigation: Supporting; Methodology: Supporting; Resources: Equal; Writing—review & editing: Equal). M.-E.A.-T. (Conceptualization: Lead; Funding acquisition: Lead; Investigation: Equal; Methodology: Supporting; Project administration: Equal; Resources: Equal; Supervision: Lead; Writing—review & editing: Equal).

## Competing interests
A patent application (application number NL2038437) related to the content of this article has been filed and is currently pending, with Technische Universiteit Delft as patent applicant, and Fan Jiang, Kui Yu, Roland Kieffer, Horia Alexandriu, and Marie-Eve Aubin-Tam as inventors. The remaining authors declare no competing interests.
