## [Peer Review File. · Communications Materials]

Thermochromic Hydrogel with High Transmittance Modulation and Fast Response for Flexible Smart Windows

Corresponding Author: Dr Marie-Eve Aubin-Tam

Version 0:

Decision Letter:

Dear Dr Aubin-Tam,

Thank you for submitting your manuscript, "Thermochromic Hydrogel with High Transmittance Modulation and Fast Response for Flexible Smart Windows", to Communications Materials. It has now been seen by 3 referees, whose comments are appended below. You will see that while they find your work of potential interest, they have raised substantial concerns that must be addressed. In light of these comments, we cannot accept the manuscript for publication, but are interested in considering a revised version that addresses these serious concerns.

In particular, the Reviewers request additional experiments to support the mechanism, as well as the monomer conversion, composition, durability, encapsulation performance, images after cycling, and outdoor temperature reduction experiments to validate energy-saving performance. The advance of this work over previous reports should also be made clearer.

We hope you will find the referees' comments useful as you decide how to proceed. Should further experimental data or analysis allow you to address these criticisms, we would be happy to look at a substantially revised manuscript. However, please bear in mind that we will be reluctant to approach the referees again in the absence of major revisions. If the revision process takes significantly longer than twelve weeks, we will be happy to reconsider your paper at a later date, as long as nothing similar has been accepted for publication at Communications Materials or published elsewhere in the meantime.

When submitting your revised manuscript, please include the following:

-A response letter with a point-by-point reply to each of the referee comments and a description of changes made. Please include the complete referee report in the response letter. Please note that the response letter must be separate to the cover letter to the editors.

-A marked-up version of the manuscript with all changes to the text in a different colored font. Please do not include tracked changes or comments. Please select the file type 'Revised Manuscript - Marked Up' when uploading the manuscript file to our online system.

-A clean version of the manuscript. Please select the file type 'Article File'.

Please use the following link to submit your revised manuscript files:

Link Redacted

Please do not hesitate to contact me if you have any questions or would like to discuss the required revisions further. Thank

you for the opportunity to review your work.

Best regards,

Dr Jet-Sing Lee
Senior Editor
Communications Materials
orcid.org/0000-0002-6740-8700

Reviewers' comments:

Reviewer #1 (Remarks to the Author):

I would like to recommend the publication of this paper in this journal, but I have several questions to communicate with the authors.

(1) Page 2, line 47: The authors stress the reflectance of "infrared light" for the opaque PNIPAM hydrogel. This may be misleading, as PNIPAM blocks light across the full solar spectrum, not just infrared.

(2) Introduction: The current progress review lacks depth. Stating that sandwiching between rigid glasses is an "ISSUE" is problematic, as this is the standard configuration for dynamic windows. Flexible substrates have niche applications but cannot replace rigid ones in most practical scenarios. Please revise to acknowledge industry norms.

(3) Page 3, after line 79: Add a paragraph explaining the design rationale for addressing the mentioned challenges. Elaborate on how the concept of EMAEMA-FMA copolymerization was conceived.

(4) Mechanical durability: Conventional PNIPAM/HPC hydrogels are brittle and shrink at temperatures above T_c . The EMAEMA-FMA hydrogel shows notable mechanical strength (as in videos/Figure 5e/Supplementary Figure S8) and no performance decay after 10 hours of thermal loading. These merits deserve stronger emphasis.

(5) Monomer conversion in Figure 1: Did UV-initiated radical copolymerization achieve 100% monomer conversion? How were reactivity ratios managed to confirm final polymer composition? Do residual monomers affect durability? Please provide validation data.

Reviewer #2 (Remarks to the Author):

In this manuscript, a flexible thermochromic smart window based on solid-state DMFM hydrogel is proposed, which features high solar transmittance modulation (70.6%) and high luminous transmittance (85.7%), fast response rate (3 s), excellent stability, and tunable transition temperature (24~39 °C) based on the water content change. The DMFM hydrogel can be encapsulated in PET films to form a lightweight flexible smart window. Compared with the limitations of high transition temperature and rigidity of traditional smart windows, DMFM/PET flexible smart window shows a wider range of application potentials, such as its application in curved glass, wearable devices and information encryption, which provides a new design concept for the development of versatility and diversified smart windows. However, the innovation of the manuscript is not highlighted, the design principle of the material is not clear, as well as the mechanism analysis needs to be further improved. Also, some formatting issues need to be corrected.

1. In the abstract section the authors do not state what the specific problem to be solved is, which leads to a lack of innovation. In addition, on page 2, line 28, "...with their optical appearance unchanged when maintained at a fixed temperature for 10 hours...", where "a fixed temperature" should be specified. The Introduction part should be improved with relevant literature, for example, <https://doi.org/10.1002/adfm.202413102>; <https://doi.org/10.1002/adma.202418372>.

2. On page 3, line 69, the authors claim that "However, most of these systems are based on liquid suspensions, requiring the thermochromic material to be sandwiched between rigid and heavy (glass) plates for smart window applications...". However, in my knowledge, there are many solid-state PNIPAM or HPC-based hydrogel materials for smart windows that have been studied and reported. The authors should highlight the innovation of this work by mainly comparing it with these works (solid-state hydrogel based materials).

3. The authors obtained DMFM hydrogel materials by copolymerizing DMAEMA monomer and FMA monomer. In this case, why is it needed to be modified with the FMA monomer, and what is the function of it, the authors should explain these in the manuscript.

4. On page 10, line 230, the authors claim that "As the thermochromic behavior of DMFM relies on the reversible phase separation between water and polymer chains, changing the proportion of water in the DMFM hydrogels is likely to tune the transition temperature...", and it has been demonstrated in subsequent experimental data that the transition temperature of DMFM hydrogels is related to the water content, however, the mechanism and reason for this should be clearly explained in the manuscript. In addition, in the manuscript the authors repeatedly emphasize that the thermochromic properties of DMFM

hydrogels are similar to those of PNIPAM hydrogels, but the transition temperatures of DMFM exhibit water content correlation whereas those of PNIPAM do not, and what is the reason for this?

5. On page 12, line 277, "Thus, the DMFM-4 smart window did not hinder the house from dissipating heat during the cooling phase, ensuring that the room did not remain excessively hot...". However, the authors have only demonstrated this by showing a significant decrease in the room temperature of the model house after turning off the IR light, which is not enough. This only shows that the whole model house exhibits good heat dissipation and cannot show that the smart window does not hinder heat dissipation. The authors need to provide more rigorous and scientific data to support this conclusion.

6. There are numerous inconsistencies between the order in the pictures and the order in which they are discussed in the manuscript, and the authors should make adjustments to make them consistent.

7. On page 9, line 213, "Fig. S6(c)" is incorrectly labeled, it should be "Fig. 3(c)".

8. There are repeated definitions of abbreviations in the manuscript, such as DMAEMA (line 74, line 97) and TLum (line 49, line 133), which are not necessary. In addition, there are cases of inconsistent formatting of some abbreviations, e.g., "UV-Vis-NIR" and "UV/Vis/NIR". The authors should carefully check and revise the full text.

9. The formatting of the references section should be consistent, especially with regard to the capitalization of article titles, e.g., Ref. [34], Ref. [40], etc.

Reviewer #3 (Remarks to the Author):

The manuscript titled "Thermochromic Hydrogel with High Transmittance Modulation and Fast Response for Flexible Smart Windows" reports the development of a hydrogel for use in flexible thermochromic devices, exhibiting rapid thermochromic response (~3 s), a tunable transition temperature range (24 °C to 39 °C), high solar modulation efficiency (70.6%), high luminous transmittance (85.7%), and excellent cycling stability over 10,000 heating/cooling cycles. However, the manuscript still presents several critical issues that need to be addressed and therefore requires major revisions before it can be considered for publication.

1. Although a solid-state hydrogel is developed in this study, it still relies on PET film encapsulation, which is fundamentally similar to conventional thermochromic hydrogel smart windows encapsulated with glass. Furthermore, flexible encapsulation typically carries a higher risk of leakage compared to rigid glass-based systems. A more detailed analysis of the encapsulation performance is therefore necessary to support the claimed advantages of this design.

2. The DMFM hydrogel exhibits impressive durability, maintaining high visible light transmittance and solar modulation performance after 10,000 thermal cycles. However, the study lacks a mechanistic investigation into the underlying factors responsible for this enhanced durability. In addition, images of the device after cyclic testing should be provided to support the durability evaluation.

3. In current research, many hydrogel materials (e.g. PNIPAM hydrogel) have shown potential for integration into flexible devices through similar encapsulation strategies. The manuscript does not clearly articulate the specific advantages that DMFM hydrogel offers over these existing materials.

4. The flexible wearable devices developed using the DMFM hydrogel appear to lack practical significance in real-world applications, which limits the applicability and impact of the research.

5. The DMFM hydrogel-based smart window is designed to aid in building thermal management and can be electrically actuated to enter the colored state. However, this mechanism introduces additional energy input, which may compromise the energy efficiency of the system. A comprehensive energy balance analysis is needed to evaluate the overall energy-saving potential of the system.

6. The thermal regulation performance of smart windows is influenced by a variety of environmental and structural factors. It is recommended to perform outdoor temperature reduction experiments to further validate the energy-saving performance of materials.

Communications Materials is committed to improving transparency in authorship. As part of our efforts in this direction, we are now requesting that all authors identified as 'corresponding author' create and link their Open Researcher and Contributor Identifier (ORCID) with their account on the Manuscript Tracking System prior to acceptance. ORCID helps the scientific community achieve unambiguous attribution of all scholarly contributions. You can create and link your ORCID from the home page of the Manuscript Tracking System by clicking on 'Modify my Springer Nature account' and following the instructions in the link below. Please also inform all co-authors that they can add their ORCIDs to their accounts and that they must do so prior to acceptance.

Version 1:

Decision Letter:

**** Please ensure you delete the link to your author homepage in this email if you wish to forward it to your coauthors ****

Dear Dr Aubin-Tam,

Thank you once again for submitting your manuscript, "Thermochromic Hydrogel with High Transmittance Modulation and Fast Response for Flexible Smart Windows," to Communications Materials. It has now been seen again by the referees, whose comments are appended below. The concerns of our reviewers have now been addressed, but there are some amendments needed before we can accept your paper.

We ask that you edit your manuscript according to the attached table. **Please read this document carefully as we will be unable to further assess your revised paper until these important points are addressed.**

Please outline all revisions made in the right-hand column and return the completed table with your updated manuscript files as a Related Manuscript file.

Please use the link below to submit your revised files:

Link Redacted

When resubmitting, please provide a marked-up manuscript with all changes highlighted, as well as a clean version of your paper.

We hope to receive this updated version of your paper within 1 week, but please let us know if you find that you need more time.

Best regards,

Dr Aldo Isidori
Senior Editor
Communications Materials

Reviewers' comments:

Reviewer #1 (Remarks to the Author):

My concerns have been fully addressed in the revised manuscript. This manuscript can be published as it is.

Reviewer #2 (Remarks to the Author):

The author has addressed all the issues I raised. I recommend accepting this manuscript.

Reviewer #3 (Remarks to the Author):

The authors have comprehensively addressed all critical concerns with high-quality experimental data, mechanistic explanations, and practical contextualization. The revised manuscript presents a well-supported study of a thermochromic hydrogel with significant advances in flexibility, durability, and tunability—with clear applications in smart windows, wearables, and active opacity control. I therefore recommend acceptance.

Version 2:

Decision Letter:

Dear Dr Aubin-Tam,

We are delighted to accept your manuscript titled "Thermochromic Hydrogel with High Transmittance Modulation and Fast Response for Flexible Smart Windows" for publication in Communications Materials. Thank you for choosing to publish your interesting work with us.

Licence to Publish and Article-Processing Charge

In approximately 7-10 business days you will receive an email with a link to choose the grant of rights necessary for publishing your paper and – if applicable – to provide payment information for your article-processing charge (APC), either via credit card or by requesting an invoice.

If needed, our Author Services team will be in touch regarding any additional information that may be required.

In order to avoid any delays, please ensure that you have emails from Springer Nature whitelisted in your mail system.

We will edit your manuscript to ensure that it conforms with our house style and send you a link to an online eProof for checking in a separate email to the publishing agreements. Please read your proof with great care to ensure that the sense has not been altered. We also suggest you discuss the proof with your co-authors, but please ensure that only one author communicates with us and that only one set of corrections is returned via the online correction in the eProof. The corresponding (or nominated) author is responsible on behalf of all co-authors for the accuracy of all content, including spelling of names and current affiliations.

To ensure prompt publication, your proofs should be returned within two working days. If there is any period within the next four weeks in which you won't be available, please nominate a co-author with whom we can correspond, and let us know their e-mail address as soon as possible.

Please note that production will not continue until the Licence to Publish and Article-Processing Charge steps are completed and your proof corrections are submitted.

Please note that your Supplementary Information files are now finalized. They will be uploaded directly to the Communications Materials website in preparation for publication of the Article. Any requests to make changes will only be considered in exceptional circumstances and will result in a delay to publication.

Acceptance of your manuscript is conditional on all authors' agreement with [our publication policies](https://www.nature.com/commsmat/editorial-policies). In particular, your manuscript must not be published elsewhere and there must be no announcement of the work in the media until the publication date. At this stage, you may wish to make your institution's press office aware of the forthcoming publication, if you wish to bring your work to the media's attention, so that they can start preparing any publicity. Please note that the paper is still under embargo until it is published in the journal. Further details of our embargo policy can be found here <http://www.nature.com/authors/policies/embargo.html>.

Publication is typically within two to three weeks of acceptance. Please note there will be no further correspondence about your publication date. When your article is published, you will receive a notification email. **If you are planning an embargoed press release or require a specific publication date, please complete our [scheduling requests form](https://forms.office.com/e/ed7NBDd08u), or contact commsproduction@springernature.com, as soon as possible after acceptance and we will endeavour to accommodate your request.** For further information on the journey of your article from acceptance to publication, please see our [Author FAQs](https://www.nature.com/documents/Author_FAQs.pdf).

If you have any questions about open-access invoicing or payment, please contact authororders@nature.com

Best regards,

John Plummer, PhD
Chief Editor
orcid.org/0000-0003-4824-8497
Communications Materials

***As a new journal, we would greatly appreciate any comments you have about your experience at Communications Materials. I hope that we have been able to meet your expectations and look forward to working with you again in the future.

We may promote your article on social media once it is published, so please feel free to send me the twitter handles of any authors or departments and we will be sure to tag them accordingly.***

Response to reviewer #1:

I would like to recommend the publication of this paper in this journal, but I have several questions to communicate with the authors.

(1) Page 2, line 47: The authors stress the reflectance of "infrared light" for the opaque PNIPAM hydrogel. This may be misleading, as PNIPAM blocks light across the full solar spectrum, not just infrared.

We thank the reviewer for this comment and suggestion. The sentence on page 2 line 47 was actually referring to the thermochromic behavior of VO₂, rather than PNIPAM hydrogel. Below T_c, the monoclinic crystalline structure of VO₂ is transparent to infrared light. Above T_c, it becomes a tetragonal crystalline structure that is reflective to infrared light [1,2]. While indeed, some modified VO₂ samples are also capable of modulating visible light transmittance. The sentence has been altered to avoid confusion: "VO₂ undergoes a reversible metal-semiconductor transition at a critical temperature (T_c) resulting in the material being transparent below the T_c while reflecting light above the T_c [4-6]."

(2) Introduction: The current progress review lacks depth. Stating that sandwiching between rigid glasses is an "ISSUE" is problematic, as this is the standard configuration for dynamic windows. Flexible substrates have niche applications but cannot replace rigid ones in most practical scenarios. Please revise to acknowledge industry norms.

We would like to clarify that it was not our intention to suggest that glass substrates pose a problem for smart window. We fully acknowledge that rigid glass is the standard in this field, owing to its excellent optical properties, durability, and widespread application in smart window systems. In the manuscript, we want to emphasize that our PET-encapsulated hydrogel devices can be easily mounted onto already existing glass windows as add-on components (Fig. 5 (a)-(c)), which serves as a complementary

strategy to broaden applications of smart window rather than replace glass substrates. As our statements might indeed lead to some confusion, we have reworded two parts of the manuscript (the added text is highlighted in yellow):

“In most polymer-based thermochromic systems, rigid glass substrates serve as the standard configuration to encapsulate the thermochromic layer, owing to their excellent optical performance, high mechanical strength and long-term stability [25]. However, flowing liquid-based hydrogel systems might be confronted with potential leakage [26].”
(page 3)

“In addition, the DMFM hydrogel can be sandwiched between lightweight and flexible PET films. Therefore, DMFM thermochromic devices can easily be installed onto existing glass surfaces of buildings or curved sunroofs in vehicles to form smart windows. While glass-based smart windows remain to be the standard configuration due to their excellent performance and durability, our approach offers a complementary strategy that broadens the scope of application scenarios.” (page 18)

(3) Page 3, after line 79: Add a paragraph explaining the design rationale for addressing the mentioned challenges. Elaborate on how the concept of EMAEMA-FMA copolymerization was conceived.

As suggested by the reviewer, we have added the following paragraph in the introduction to explain our design rationale:

“In this study, we address these challenges in thermochromic smart window design by developing a solid-state, and flexible material with thermal stability that exhibits a rapid and tunable thermochromic response within the human comfort temperature range. The material should ideally possess the advantages of polymer-based systems, such as low transition temperatures and high ΔT_{Sol} , as well as the structural integrity of solid-state devices. Poly(DMAEMA) hydrogel is a thermochromic material with transition

temperature around 50 °C, tunable thermal responsiveness, and facile curing conditions [38], presenting huge potential and value for optimization. To further reduce the transition temperature and increase the response rate for practical applications, we copolymerized DMAEMA with hydrophobic fluorinated monomers. Owing to weakened water bonding and enhanced water repellency by the hydrophobic moieties, the co-polymerized material experienced a more readily triggered phase separation, leading to lower transition temperature and faster opacity change.”

(4) Mechanical durability: Conventional PNIPAM/HPC hydrogels are brittle and shrink at temperatures above T_c . The EMAEMA-FMA hydrogel shows notable mechanical strength (as in videos/Figure 5e/Supplementary Figure S8) and no performance decay after 10 hours of thermal loading. These merits deserve stronger emphasis.

We thank the reviewer for this suggestion. These are great points to emphasize. Please find the rewritten and supplemented parts below:

“The responsiveness of the DMFM-4 hydrogel device to both temperature changes and mechanical stimuli is elegantly evidenced in Video S1, where there is a visible change in turbidity and reversible deformation in shape upon being held and bent by hand.” (page 5)

“Notably, the material still displayed thermochromic behavior when stretched by 500%, without showing obvious shrinkage or embrittlement above the transition temperature, demonstrating great mechanical and thermal stability.” (page 13)

“The DMFM hydrogel exhibits great mechanical flexibility, thermal stability, and durable thermochromic performance, highlighting its potential for long-term application in smart window systems.” (page 18)

(5) Monomer conversion in Figure 1: Did UV-initiated radical copolymerization

achieve 100% monomer conversion? How were reactivity ratios managed to confirm final polymer composition? Do residual monomers affect durability? Please provide validation data.

To qualitatively assess the degree of monomer conversion, we fabricated devices with additional photoinitiator (2.5 wt%) and tested them with FT-IR. We expect that the addition of excessive photoinitiator would ensure higher monomer conversion, but with the risk of yellowing of the hydrogel. As shown in Fig. R1(a), the FT-IR spectra show that the C=C stretching band at around 1647 cm^{-1} and the C=C twisting bands from 900 to 800 cm^{-1} of DMFM-4 hydrogel experienced a decrease after being cured with excessive photoinitiator (Fig. R1(b, c)). This phenomenon indicated further conversion of monomer, confirming that not all monomer is converted in the original formulation.

Fig. R1 FT-IR spectra of the DMFM-4 precursor solution, DMFM-4 hydrogel, and the corresponding hydrogel cured with excessive amount of photoinitiator, highlighting key spectral regions: (a) $4000\text{--}600\text{ cm}^{-1}$; (b) $1675\text{--}1600\text{ cm}^{-1}$; (c) $925\text{--}800\text{ cm}^{-1}$.

Further, we know that the uncured DMFM-4 precursor solution can also perform

reversible opacity change under variations of temperature. Therefore, we consider that the presence of monomer in the DMFM hydrogel would not greatly affect the thermochromism property and durability. We have modified the manuscript as follows: “The uncured DMFM-4 precursor also presents a reversible change in opacity in response to temperature variations (Fig. S2).” (page 5)

Fig. S2 Photographs of uncured DMFM-4 precursor solution (a) at transparent state at $T = 20\text{ }^{\circ}\text{C}$; and (b) at turbid state at $T = 30\text{ }^{\circ}\text{C}$ in a 20 mL glass vial in standing positions (left) and in horizontal positions (right).

Concerning the polymer composition, we acknowledge that the initial ratios of reactants may not directly reflect the final polymer composition. We want to clarify that our study intentionally reports mixing ratios rather than final polymer composition.

References

- [1] Cui Y., Ke Y., Liu C., et al. Thermochromic VO_2 for energy-efficient smart windows. *Joule*, **2018**, 2(9): 1707-1746.
- [2] Zhou J., Gao Y., Zhang Z., et al. VO_2 thermochromic smart window for energy savings and generation. *Sci. Rep.*, **2013**, 3(1): 3029.

Response to reviewer #2:

In this manuscript, a flexible thermochromic smart window based on solid-state DMFM hydrogel is proposed, which features high solar transmittance modulation (70.6%) and high luminous transmittance (85.7%), fast response rate (3 s), excellent stability, and tunable transition temperature (24~39 °C) based on the water content change. The DMFM hydrogel can be encapsulated in PET films to form a lightweight flexible smart window. Compared with the limitations of high transition temperature and rigidity of traditional smart windows, DMFM/PET flexible smart window shows a wider range of application potentials, such as its application in curved glass, wearable devices and information encryption, which provides a new design concept for the development of versatility and diversified smart windows. However, the innovation of the manuscript is not highlighted, the design principle of the material is not clear, as well as the mechanism analysis needs to be further improved. Also, some formatting issues need to be corrected.

1. In the abstract section the authors do not state what the specific problem to be solved is, which leads to a lack of innovation.

We thank the reviewer for their time and insights. Please find the revised abstract hereafter:

“Growing environmental concerns are increasing the demand for strategies to reduce urban energy consumption. One practical solution for residential and office buildings is thermochromic smart windows, which offer passive temperature management by autonomously regulating sunlight. Despite recent progress, current technologies still face challenges in achieving the thermal durability and mechanical robustness necessary for long-term use, combined with a rapid transition below 30°C. Here we report a thermochromic hydrogel assembled from poly(N,N-dimethylaminoethyl methacrylate) and 2,2,2-trifluoroethyl methacrylate that produces flexible films on a

large scale. This hydrogel rapidly ($\Delta t \approx 3$ s) and reversibly becomes turbid above a transition temperature (tunable from 24°C to 39°C) which notably spans the human comfort zone. The film's high modulation of solar transmittance (70.6%) and luminous transmittance (85.7%) enables efficient sunlight screening in hot weather and clear vision in cool weather. Such 'smart windows' remain stable for over 10,000 heating/cooling cycles and their optical appearance remains unchanged when maintained at a fixed temperature of 35°C for 10 hours. Furthermore, this effect is still present even when mechanically stretched with 500% strain. These combined features indicate the hydrogel suitability for applications ranging from heat-modulating smart windows (architectural, automotive, etc.) to passive temperature indicators and even wearables."

In addition, on page 2, line 28, "...with their optical appearance unchanged when maintained at a fixed temperature for 10 hours...", where "a fixed temperature" should be specified.

We thank the reviewer for the suggestion and for pointing the missing of temperature in abstract. Although the temperature used for 10-hour measurement had been specified on page 9, line 212, it was not included in abstract. We have now included it in the abstract (see point 1 above).

The Introduction part should be improved with relevant literature, for example, <https://doi.org/10.1002/adfm.202413102>; <https://doi.org/10.1002/adma.202418372>.

We thank the reviewer for the suggestion and provided literature. We have carefully read the recommended references and agree that they provide valuable context related to our work. We have supplemented relevant literature in the introduction. Please find hereafter the supplemented literature regarding relevant solid-state hydrogels:

- [27] Wang W., Wang K., Cheng Y., et al. Bidirectional temperature-responsive thermochromic hydrogels with adjustable light transmission interval for smart windows. *Adv. Funct. Mater.*, **2024**: 2413102.
- [28] Wu C., Cheng Y., Wang K., et al. Temperature-mediated controllable adhesive hydrogels with remarkable wet adhesion properties based on dynamic interchain interactions. *Adv. Funct. Mater.*, **2025**, 35(19): 2423099.
- [29] Wang K., Chen G., Weng S., et al. Thermo-responsive poly (N-isopropylacrylamide)/hydroxypropylmethyl cellulose hydrogel with high luminous transmittance and solar modulation for smart windows. *ACS Appl. Mater. Interfaces*, **2023**, 15(3): 4385-4397.
- [30] Sun M., Sun H., Wei R., et al. Energy-efficient smart window based on a thermochromic hydrogel with adjustable critical response temperature and high solar modulation ability. *Gels*, **2024**, 10(8): 494.
- [31] Wei G., Yang D., Zhang T., et al. Thermal-responsive PNIPAM-acrylic/Ag NRs hybrid hydrogel with atmospheric window full-wavelength thermal management for smart windows. *Sol. Energy Mater. Sol. Cells*, **2020**, 206: 110336.
- [32] Wang K., Liu S., Yu J., et al. Hofmeister effect-enhanced, nanoparticle-shielded, thermally stable hydrogels for anti-UV, fast-response, and all-day-modulated smart windows. *Adv. Mater.*, **2025**, 37(14): 2418372.
- [33] Guo N., Liu S., Chen C., et al. Outdoor adaptive temperature control based on a thermochromic hydrogel by regulating solar heating. *Sol. Energy*, **2024**, 270: 112405.”

2. On page 3, line 69, the authors claim that “However, most of these systems are based on liquid suspensions, requiring the thermochromic material to be sandwiched between rigid and heavy (glass) plates for smart window applications...”. However, in my knowledge, there are many solid-state PNIPAM or HPC-based hydrogel materials for smart windows that have been studied and reported. The authors should highlight the innovation of this work by mainly comparing it with these works (solid-state hydrogel based materials).

In the revised manuscript, we have highlighted the novelty of our work and compared it to these works:

“In most polymer-based thermochromic systems, rigid glass substrates serve as the standard configuration to encapsulate the thermochromic layer, owing to their excellent optical performance, high mechanical strength and long-term stability [25]. However, such flowing liquid-based hydrogel systems might be confronted with potential leakage [26]. The PNIPAM hydrogel can also be prepared in the solid state and improved through various methods, such as copolymerizing with other monomers (acrylamide, acrylic acid) [27,28], adding cellulose derivatives (hydroxypropylmethylcellulose (HPMC), HPC) [29,30], and doping the silver nanorods [31]. Other organic solid-state thermochromic materials have also been studied, including HPMC-based hydrogel [32], HPC composite hydrogel [33], ionogels [34] and N,N-dimethylaminoethyl methacrylate-based (DMAEMA) hydrogels [35,36]. These solid state thermochromic systems exhibit promising heat modulation and anti-leakage properties, but they likewise show limitations such as a slow response or a transition temperature outside the temperature comfort zone of the human body, i.e. between 20 – 30 °C [37] (see Table S1 Supplementary Information for a summary of the performance of existing technology). Moreover, the durability and thermal stability of many smart window systems have not been fully explored and optimized.

In this study, we address these challenges in thermochromic smart window design by developing a solid-state, and flexible material with thermal stability that exhibits a rapid and tunable thermochromic response within the human comfort temperature range.”

(pages 3-4)

Table S1 The performances of current state-of-the-art thermochromic smart window systems.

ΔT_{solar}	T_{lim}	Transition temperature	Response time	Cycles	Solid State	Flexibility	Tunability	Materials	Fabrication process	Ref
7.50%	45.60%	68 °C	-	-	Yes	No	No	VO ₂ film	Spin-coating and annealing	[2]
3.3-7.6%	47.5-48.9%	44.9-55.0 °C	-	-	Yes	No	Yes	Al-doped VO ₂ film	Magnetron sputtering	[3]
1.0-5.0%	27.7-40.6%	34.4-50.7 °C	-	-	Yes	No	Yes	Mg-doped VO ₂ film	Magnetron sputtering	[4]
3.3-7.4%	48.2-63.7%	26.0-57.0 °C	-	-	Yes	No	Yes	W-doped VO ₂ film	Pulsed laser deposition	[5]
14.10%	60.40%	64.3 °C	-	-	Yes	Yes	No	Zr-doped VO ₂ NPs in PU	Hydrothermal synthesis and curing	[6]
4.9-12.3%	48.6-58.4%	28.6-55.9 °C	-	-	Yes	Yes	Yes	Zr-W-doped VO ₂ NPs in PU	Hydrothermal synthesis and curing	[6]
17.20%	53.00%	66.9 °C	-	-	Yes	Yes	No	Ti-doped VO ₂ NPs in PU	Hydrothermal synthesis and curing	[7]
5.6-11.0%	45.3-54.2%	54-67 °C	-	-	Yes	Yes	Yes	Mg-doped VO ₂ NPs in PU	Hydrothermal synthesis and curing	[8]
10.0-18.6%	63.3-82.5%	56.8-65.2 °C	-	-	Yes	Yes	Yes	W ₁₈ O ₄₉ -doped VO ₂ NPs in PVP	Solvothermal synthesis and curing	[9]
37.70%	35.20%	65 °C	-	100	Yes	Yes	No	VO ₂ NPs in PDMS	Mixing and curing	[10]
49.60%	85.80%	-32 °C	-	20	No	No	No	PNIPAM	Mixing and curing	[11]
34.70%	62.60%	35 °C	-	-	No	No	No	PNIPAM/VO ₂	Mixing and curing	[12]
73.50%	88.00%	32.5 °C	17 s	200	Yes	Yes	No	PNIPAM microgel in Si/Al-gel	Mixing and curing	[13]
69.65%	87.37%	27.2 °C	3 mins	100	No	No	No	PNIPAM/KCA/Na ₂ SiO ₃ suspension	Mixing and curing	[14]
68.10%	-90%	32.5 °C	-	100	No	No	No	PNIPAM liquid	Polymerization and freeze-drying	[15]
81.30%	87.20%	32 °C	-	1,000	No	No	No	PNIPAM-AEMA microgel	Mixing and curing	[16]
60.80%	89.20%	19.1 - 32.7 °C	9.6 s	10	No	No	Yes	PNIPAM GW solutions	Mixing and curing	[17]
47.50%	90.10%	10 - 44 °C	184 s	100	No	No	Yes	HPC/PAAc	Mixing	[18]
69.50%	84.40%	33.1 - 47.8 °C	20 s	50	Yes	No	Yes	P(AAm-co-AA)/NIPAM/AAm hydrogel	Mixing and curing	[19]
81.52%	90.82%	32 °C	-	100	Yes	No	No	PNIPAM/HPMC hydrogel	Mixing and curing	[20]
87.50%	71.20%	24.1 - 33.2 °C	40 s	100	Yes	No	Yes	HBPEC/PNIPAM hydrogel	Mixing and curing	[21]
61.36%	59.24%	32.9 °C	-	20	Yes	No	No	PNIPAm-acrylic/Ag NRs hybrid hydrogel	Mixing and curing	[22]
66.90%	51.20%	22 - 50.2 °C	30 s	100	Yes	No	Yes	HPMC/PDMAA hydrogel	Mixing and curing	[23]
-	-	25 - 45 °C	-	-	Yes	Yes	Yes	HPC/NaCl composite hydrogel	Mixing and curing	[24]
66.80%	87.00%	20 - 100 °C	3 mins	5,000	Yes	Yes	Yes	PU gel with ionic liquids	Mixing and curing	[25]
-	-	47 - 52.5 °C	-	-	Yes	Yes	Yes	P(DMAEMA-co-COU) hydrogel	Mixing and curing	[26]
-	76.40%	40 - 52 °C	28 s	500	Yes	Yes	Yes	P(DMAEMA-co-TSPM) hydrogel	Mixing and curing	[27]
70.64%	85.67%	24 - 39 °C	3 s	10,000	Yes	Yes	Yes	P(DMAEMA-FMA) hydrogel	Mixing and curing	This work

3. The authors obtained DMFM hydrogel materials by copolymerizing DMAEMA monomer and FMA monomer. In this case, why is it needed to be modified with the FMA monomer, and what is the function of it, the authors should explain these in the manuscript.

The FMA monomer is copolymerized to increase the hydrophobicity of hydrogel, which leads to lower transition temperature and faster thermochromic response. We have supplemented statements in the manuscript to emphasize the importance of FMA.

“The material should ideally possess the advantages of polymer-based systems, such as low transition temperatures and high ΔT_{sol} , as well as the structural integrity of solid-state devices. Poly(DMAEMA) hydrogel is a thermochromic material with transition temperature around 50 °C, tunable thermal responsiveness, and facile curing conditions [38], presenting huge potential and value for optimization. To further reduce the transition temperature and increase the response rate for practical applications, we copolymerized DMAEMA with hydrophobic fluorinated monomers. Owing to weakened water bonding and enhanced water repellency by the hydrophobic moieties, the co-polymerized material experienced a more readily triggered phase separation, leading to lower transition temperature and faster opacity change.” (page 4)

“The lower transition temperature in comparison to poly(DMAEMA) [39] is attributed to the copolymerization with FMA, where the hydrophobic trifluoromethyl groups of FMA promote a more readily triggered phase separation.” (page 5)

4. On page 10, line 230, the authors claim that “As the thermochromic behavior of DMFM relies on the reversible phase separation between water and polymer chains, changing the proportion of water in the DMFM hydrogels is likely to tune the transition temperature...”, and it has been demonstrated in subsequent experimental data that the transition temperature of DMFM hydrogels is related to the water content, however, the mechanism and reason for this should be clearly explained in the manuscript.

In addition, in the manuscript the authors repeatedly emphasize that the thermochromic properties of DMFM hydrogels are similar to those of PNIPAM hydrogels, but the transition temperatures of DMFM exhibit water content correlation whereas those of PNIPAM do not, and what is the reason for this?

We thank the reviewer for their comments and suggestion. So far, we hypothesize that the combination of amine groups in DMAEMA and trifluoromethyl groups in FMA lead to the tunable transition temperature in the DMFM hydrogel. We describe this further in our manuscript (see changes hereafter).

We indeed pointed out that the PNIPAM hydrogels and our DMFM hydrogel shared similar thermochromic mechanism based on water-polymer phase separation, while these two materials are still different due to their different functional groups and water-polymer interactions. The thermochromic property of PNIPAM hydrogel is based on the hydrophilic amide groups and hydrophobic isopropyl groups. As opposed to the -CONH- group of PNIPAM, the -N(CH₃)₂ group of poly(DMAEMA) can be protonated and become charged depending on the affinity pH values and salt concentrations [1,2], endowing poly(DMAMEA) hydrogel with tunable transition temperature [3,4]. Further, the incorporation of FMA introduces highly hydrophobic trifluoromethyl groups into the DMFM copolymer [5]. These fluorine-containing groups exhibit much higher hydrophobicity than alkyl groups [6,7], such as the methyl/ethyl groups on DMAEMA or isopropyl groups on PNIPAM, therefore enhancing the sensitivity upon water concentration of DMFM hydrogel.

We have modified our manuscript as follows:

“As opposed to PNIPAM, the transition temperatures of DMFM hydrogels relates to the water concentration. We hypothesize that this originates from the protonation of tertiary amine groups and corresponding change of charge density in the hydrogel [50,51], which affects the hydrogen bonding network and polymer hydrophilicity

[42,52]. In addition, the polymer chains with hydrophobic trifluoromethyl groups could aggregate more easily in a water-rich hydrogel [53].” (page 11)

5. On page 12, line 277, “Thus, the DMFM-4 smart window did not hinder the house from dissipating heat during the cooling phase, ensuring that the room did not remain excessively hot...”. However, the authors have only demonstrated this by showing a significant decrease in the room temperature of the model house after turning off the IR light, which is not enough. This only shows that the whole model house exhibits good heat dissipation and cannot show that the smart window does not hinder heat dissipation. The authors need to provide more rigorous and scientific data to support this conclusion.

We thank the reviewer for their comment. As this work does not aim to investigate cooling performance at lower temperatures, but rather the opacity change and heat-blocking behavior of materials under elevated temperatures, we agree that the discussion of cooling might be unnecessary and also confusing. We have revised the text accordingly to avoid misunderstanding:

“After the light was turned off, the inside temperature dropped down to ~ 24 °C within a similar time period of 15 minutes for all test groups.” (page 13)

6. There are numerous inconsistencies between the order in the pictures and the order in which they are discussed in the manuscript, and the authors should make adjustments to make them consistent.

We thank the reviewer for pointing out these inconsistencies in the manuscript. We have carefully reviewed the manuscript and revised both the text and figure order to ensure consistency and clarity.

7. On page 9, line 213, “Fig. S6(c)” is incorrectly labeled, it should be “Fig. 3(c)”.

We thank the reviewer for catching this. We have corrected the manuscript.

8. There are repeated definitions of abbreviations in the manuscript, such as DMAEMA (line 74, line 97) and TLum (line 49, line 133), which are not necessary. In addition, there are cases of inconsistent formatting of some abbreviations, e.g., “UV-Vis-NIR” and “UV/Vis/NIR”. The authors should carefully check and revise the full text.

We thank the reviewer for this comment. We have revised the definitions and abbreviations in the manuscript.

9. The formatting of the references section should be consistent, especially with regard to the capitalization of article titles, e.g., Ref. [34], Ref. [40], etc.

We thank the reviewer for pointing this out. We have corrected the format of the references.

References

- [1] Cotanda P., Wright D. B., Tyler M., et al. A comparative study of the stimuli-responsive properties of DMAEA and DMAEMA containing polymers. *J. Polym. Sci., Part A: Polym. Chem.*, **2013**, 51(16): 3333-3338.
- [2] Karanikolopoulos N., Zamurovic M., Pitsikalis M., et al. Poly (dl-lactide)-b-poly (N,N-dimethylamino-2-ethyl methacrylate): synthesis, characterization, micellization behavior in aqueous solutions, and encapsulation of the hydrophobic drug dipyridamole. *Biomacromolecules*, **2010**, 11(2): 430-438.
- [3] Plamper F. A., Ballauff M., Müller A. H. E. Tuning the thermoresponsiveness of weak polyelectrolytes by pH and light: lower and upper critical-solution temperature of poly (N, N-dimethylaminoethyl methacrylate). *J. Am. Chem. Soc.*, **2007**, 129(47): 14538-14539.
- [4] Fournier D., Hoogenboom R., Thijs H. M. L., et al. Tunable pH-and temperature-sensitive copolymer libraries by reversible addition-fragmentation chain transfer

- copolymerizations of methacrylates. *Macromolecules*, **2007**, 40(4): 915-920.
- [5] Biffinger J. C., Kim H. W., DiMagno S. G. The polar hydrophobicity of fluorinated compounds. *ChemBioChem*, **2004**, 5(5): 622-627.
- [6] Muller N. When is a trifluoromethyl group more lipophilic than a methyl group? Partition coefficients and selected chemical shifts of aliphatic alcohols and trifluoroalcohols. *J. Pharm. Sci.*, **1986**, 75(10): 987-991.
- [7] Jeffries B., Wang Z., Graton J., et al. Reducing the lipophilicity of perfluoroalkyl groups by CF₂-F/CF₂-Me or CF₃/CH₃ exchange. *J. Med. Chem.*, **2018**, 61(23): 10602-10618.

Response to reviewer #3:

The manuscript titled “Thermochromic Hydrogel with High Transmittance Modulation and Fast Response for Flexible Smart Windows” reports the development of a hydrogel for use in flexible thermochromic devices, exhibiting rapid thermochromic response (~3 s), a tunable transition temperature range (24 °C to 39 °C), high solar modulation efficiency (70.6%), high luminous transmittance (85.7%), and excellent cycling stability over 10,000 heating/cooling cycles. However, the manuscript still presents several critical issues that need to be addressed and therefore requires major revisions before it can be considered for publication.

1. Although a solid-state hydrogel is developed in this study, it still relies on PET film encapsulation, which is fundamentally similar to conventional thermochromic hydrogel smart windows encapsulated with glass.

We thank the reviewer for their time and comments. We fully acknowledge that rigid glass is the standard in this field, owing to its excellent optical properties, durability, and widespread application in smart window systems. In our study, we explored PET as an alternative encapsulation material primarily to demonstrate its potential advantages in specific application scenarios. As opposed to conventional smart windows encapsulated with glass, PET-based devices offer the benefit of being lightweight, which allows for direct integration onto existing glass surfaces (shown in Fig. 5(a)-(c)), potentially reducing installation time and cost. The flexibility of the PET device also opens up new application possibilities, such as integration into non-traditional surfaces (curved glass panels, car windows), and portable electronic devices. We have modified the manuscript to clarify this:

“While glass-based smart windows remain to be the standard configuration due to their excellent performance and durability, our approach offers a complementary strategy that broadens the scope of application scenarios.” (page 18)

Additionally, we would like to clarify that the thermochromic property of the hydrogel is not necessarily dependent on PET encapsulation. As demonstrated in Fig. S11(b), the DMFM hydrogel retains its thermochromic performance while in a self-standing state and even when being stretched. This observation suggests that the intrinsic material properties of the DMFM hydrogel play a key role in the thermochromic behavior.

Furthermore, flexible encapsulation typically carries a higher risk of leakage compared to rigid glass-based systems. A more detailed analysis of the encapsulation performance is therefore necessary to support the claimed advantages of this design.

We appreciate the reviewer's concern regarding the potential risk of leakage associated with flexible encapsulation. In our case, however, the system is not liquid-based, which significantly reduces such risks. As shown in Video S1, the device remains intact and functional even under bending conditions, indicating good structural stability.

To demonstrate this, we have performed additional tests. To evaluate the anti-leakage performance, a DMFM-4 hydrogel PET device (dimensions: 5 cm * 5 cm * 0.5 mm) underwent 100 times of falling from a height of 1 meter and 100 cycles of 90° bending. The appearance and weight of the device were recorded before, during, and after these tests.

The following text and new experimental data were added to describe these experiments:

“We found that after multiple drops from 1-meter height and subsequent bending to 90°, no visible leakage or structural damage was observed in the PET-encapsulated DMFM-4 hydrogel device (Fig. S12(a)). The unchanged weight of the device after dropping and bending further confirmed the anti-leakage property (Fig. S12(b)), owing to the encapsulation with PET and mechanical strength of DMFM hydrogel.” (page 14)

Fig. S12 (a) Photographs of PET-encapsulated DMFM-4 hydrogel device (dimensions: 5 cm * 5 cm * 0.5 mm) in as-prepared state, after 50 and 100 drops from a height of 1 meter, after being bended 50 and 100 times to 90°; (b) the corresponding weight measurements of PET-encapsulated DMFM-4 hydrogel device under each condition.

2. The DMFM hydrogel exhibits impressive durability, maintaining high visible light transmittance and solar modulation performance after 10,000 thermal cycles. However, the study lacks a mechanistic investigation into the underlying factors responsible for this enhanced durability.

We thank the reviewer for this comment. While 10,000 thermal cycles under comparable conditions have not yet been systematically reported by many previous studies, we acknowledge that this does not necessarily imply that other thermochromic systems are incapable of achieving similar or better performance. Therefore, we prefer not to describe the observed durability as “enhanced” since it remains possible that other systems could demonstrate comparable durability.

The following text was added to describe factors contributing to durability:

“The combination of good mechanical properties and thermal stability over 10,000 cycles indicate an excellent durability for long-term operation. This durability could be attributed to several factors: Firstly, the encapsulation with PET effectively protects the

hydrogel from direct mechanical damage, leading to longer life time during thermal cycles. Secondly, the polymerized solid-state hydrogel exhibits good mechanical strength and structural robustness, which helps to maintain its structure and performance under repeated thermal stress. We found that after multiple drops from 1-meter height and subsequent bending to 90°, no visible leakage or structural damage was observed in the PET-encapsulated DMFM-4 hydrogel device (Fig. S12(a)). The unchanged weight of the device after dropping and bending further confirmed the anti-leakage property (Fig. S12(b)), owing to the encapsulation with PET and mechanical strength of DMFM hydrogel. Thirdly, we expect that the crosslinking of hydrogel helps to construct a more uniform and well-aligned polymer chain network, further enhancing the reversibility of phase separation the DMFM hydrogel.” (pages 13-14)

In addition, images of the device after cyclic testing should be provided to support the durability evaluation.

We have supplemented images of the device after 10,000 heating-cooling cycles to further support the durability data. These are included in the Supporting Information. The following text and new experimental data were added to the manuscript:

The DMFM-4 device remained transparent and homogeneous after 10,000 cycles and could still demonstrate opacity change under higher temperature (Fig. S8).” (page 10)

Fig. S8 Optical images of DMFM-4 hydrogel device (4 cm * 4 cm *0.5 mm) before and after 10,000 heating-cooling cycles. (a) under transparent state at 20 °C before 10,000 cycles; (b) under turbid state at 40 °C before 10,000 cycles; (c) under transparent state at 20 °C after 10,000 cycles; (d) under turbid state at 40 °C after 10,000 cycles.

3. In current research, many hydrogel materials (e.g. PNIPAM hydrogel) have shown potential for integration into flexible devices through similar encapsulation strategies. The manuscript does not clearly articulate the specific advantages that DMFM hydrogel offers over these existing materials.

We thank the reviewer for this comment. Firstly, we would like to clarify that the PET encapsulation serves as protection of DMFM hydrogel against water loss and environmental damage to improve stability. Secondly, we consider the copolymerization of DMAEMA and hydrophobic FMA can lead to rapid response and lower transition temperature of hydrogel. Thirdly, compared with some reported thermochromic hydrogels which are liquid or brittle solid-state [1,2], our DMFM hydrogel is flexible and maintains its shape above transition temperature (Fig. S11).

Finally, the DMFM hydrogel also simultaneously exhibits several advantages, including tunable transition temperatures, high luminous transmittance (~85%), comparable solar modulation (~70%), fast response ~3 seconds, and great thermal durability. Please find the supplemented parts that articulate the advantages of DMFM hydrogel hereafter:

“By encapsulating this solid-state hydrogel with notable mechanical strength in a soft and transparent substrate, such as polyethylene terephthalate (PET), it can be fabricated into a flexible thermochromic device that is capable of operating for more than 10,000 heating/cooling cycles and 10 hours of continuous operation without visible performance decay.” (page 4)

“The lower transition temperature in comparison to poly(DMAEMA) is attributed to the copolymerization with FMA, where the hydrophobic trifluoromethyl groups of FMA promote a more readily triggered phase separation.” (page 5)

“Notably, the material still displayed thermochromic behavior when stretched by 500%, without showing obvious shrinkage or embrittlement above the transition temperature.” (page 13)

See also the text that we have added to the manuscript in response to points 1 and 2.

4. The flexible wearable devices developed using the DMFM hydrogel appear to lack practical significance in real-world applications, which limits the applicability and impact of the research.

In the revised manuscript, we further clarify the applicability of the DMFM hydrogel based thermochromic wearables:

“The hat brim becomes opaque under high temperature and blocks sunlight from the

wearer, but turns transparent under cooler or low-light conditions to enhance upward visibility when shading is unnecessary. This demonstration is a simple and battery-free solution for passive sun shading and real-time temperature indication, which can be incorporated into outdoor clothing, sport gears, fashion-related accessories, or shades of outdoor terraces.” (page 14)

We also wish to point that similar wearable applications have been explored in previous studies, such as clothing made of thermochromic textiles for wearable temperature sensors [3,4], skirts fabricated by thermochromic fibers that display different patterns for aesthetic purposes [5,6], and thermochromic hydrogel wearables for sport signal monitoring and body temperature management [7]. Our work similarly aims to contribute to the area of wearables by introducing a new thermochroic material with both practical and visual functionality.

5. The DMFM hydrogel-based smart window is designed to aid in building thermal management and can be electrically actuated to enter the colored state. However, this mechanism introduces additional energy input, which may compromise the energy efficiency of the system. A comprehensive energy balance analysis is needed to evaluate the overall energy-saving potential of the system.

As mentioned on page 14, line 312, the electrically controlled hydrogel device is not intended for heat modulation or energy saving, but rather as a way to demonstrate that the DMFM hydrogel can become opaque “on demand” through electrical input if needed. This may be useful in specific application scenarios, for privacy modulation, actively switchable protective shades against strong sunlight, and aesthetic or interactive elements in architectural design.

For completeness, we have also calculated the electrical power consumption of the electrically driven DMFM hydrogel device, as presented in the revised manuscript:

“The measured electrical resistance of this device is 87.9 Ω .” (page 20)

“this bendable smart window stayed transparent in the OFF state (0 V), and turned turbid uniformly in the whole device area after switching to the ON state (7.0 V), using 0.56 W of electrical power.” (page 16)

6. The thermal regulation performance of smart windows is influenced by a variety of environmental and structural factors. It is recommended to perform outdoor temperature reduction experiments to further validate the energy-saving performance of materials.

In response to the reviewer’s suggestion, we have further evaluated the material with outdoor experiments to better characterize its performance, supplementing the analysis of transmittance modulation (Fig. 2(a)-(b)) and indoor model house simulation tests (Fig. 5(a)-(c)). Here is the added text and new experimental data:

“Tests were also performed on a model house outdoors on a sunny day, confirming that DMFM-4 smart windows led to reduced heating compared to glass windows (Fig. S10).” (page 13)

Fig. S10 (a) Pictures of outdoor environment and simulation tests; (b) temperature profiles of temperature sensors in simulated houses (10 cm * 7.5 cm * 5 cm) with glass window or with

DMFM-4 hydrogel device (8 cm * 5.5 cm * 0.5 mm) on the glass window, or in ambient environment exposed to outdoor solar radiation.

Reference

- [1] Li X. H., Liu C., Feng S. P., et al. Broadband light management with thermochromic hydrogel microparticles for smart windows. *Joule*, **2019**, 3(1): 290-302.
- [2] Sun M., Sun H., Wei R., et al. Energy-efficient smart window based on a thermochromic hydrogel with adjustable critical response temperature and high solar modulation ability. *Gels*, **2024**, 10(8): 494.
- [3] Lim T., Seo H. S., Yang J., et al. Reversible thermochromic fibers with excellent elasticity and hydrophobicity for wearable temperature sensors. *RSC Adv.*, **2024**, 14(9): 6156-6164.
- [4] Lee C., Tan J., Tan J. J., et al. Intelligent thermochromic heating e-textile for personalized temperature control in healthcare. *ACS Appl. Mater. Interfaces*, **2025**, 17(3): 5515-5526.
- [5] Yang L., Meng J., Yu L., et al. Reversible dual-responsive color-changing fabric based on thermochromic microcapsules for textile fashion and intelligent monitoring. *Dyes Pigm.*, **2024**, 231: 112397.
- [6] Zhan L., Xu W., Hu Z., et al. Full-color “off-on” thermochromic fluorescent fibers for customizable smart wearable displays in personal health monitoring. *Small*, **2024**, 20(29): 2310762.
- [7] Xie L., Wang X., Zou X., et al. Engineering self-adaptive multi-response thermochromic hydrogel for energy-saving smart windows and wearable temperature-sensing. *Small*, **2023**, 19(52): 2304321.